JCB Journal of Cell Biology

# Senescent cells suppress macrophage-mediated corpse removal via upregulation of the CD47-QPCT/L axis

Daniela Schloesser[1]*, Laura Lindenthal[2]*, Julia Sauer[1], Kyoung-Jin Chung[3], Triantafyllos Chavakis[3], Eva Griesser[1], Praveen Baskaran[1], Ulrike Maier-Habelsberger[1], Katrin Fundel-Clemens[1], Ines Schlotthauer[1], Carolin Kirsten Watson[1], Lee Kim Swee[1], Frederik Igney[1], John Edward Park[1], Markus S. Huber-Lang[4], Matthew-James Thomas[1], Karim Christian El Kasmi[1], and Peter J. Murray[2]

Progressive accrual of senescent cells in aging and chronic diseases is associated with detrimental effects in tissue homeostasis. We found that senescent fibroblasts and epithelia were not only refractory to macrophage-mediated engulfment and removal, but they also paralyzed the ability of macrophages to remove bystander apoptotic corpses. Senescent cell-mediated efferocytosis suppression (SCES) was independent of the senescence-associated secretory phenotype (SASP) but instead required direct contact between macrophages and senescent cells. SCES involved augmented senescent cell expression of CD47 coinciding with increased CD47-modifying enzymes QPCT/L. SCES was reversible by interfering with the SIRPα-CD47-SHP-1 axis or QPCT/L activity. While CD47 expression increased in human and mouse senescent cells in vitro and in vivo, another ITIM-containing protein, CD24, contributed to SCES specifically in human epithelial senescent cells where it compensated for genetic deficiency in CD47. Thus, CD47 and CD24 link the pathogenic effects of senescent cells to homeostatic macrophage functions, such as efferocytosis, which we hypothesize must occur efficiently to maintain tissue homeostasis.

## Introduction

A key question in understanding the processes of aging, and many diseases linked to aging, concerns the contributions of diseased and defective cells to tissue-level and organismal health. Senescent cells are "defective" cells that progressively accrue in aging (López-Otín et al., 2013), and in numerous chronic diseases (Coppé et al., 2010), including lung and liver fibrosis, atherosclerosis, and obesity (Childs et al., 2017). A defined negative effect of senescent cell accrual is that following permanent exit from the cell cycle, they form reservoirs of cells with DNA damage and aneuploidy; a fraction of these cells have the ability to escape the cell cycle block through acquisition of potentially oncogenic mutations, leading to an age-related risk of malignancy (Sharpless and Sherr, 2015). A second detrimental effect of senescent cells is their potential to modify tissue environments by the secretion of a cocktail of pro-inflammatory and pro-proliferative mediators, termed the senescence-associated secretory phenotype (SASP), which recruit and activate immune cells, such as macrophages, and perpetuate local inflammation (Coppé et al., 2010). However, beyond the potential role of the SASP, the mechanisms through which senescent cells drive disease pathology and disrupt or antagonize local tissue environments is uncharted. Genetic and pharmacological approaches in modeled in vivo systems to reduce the number of senescent cells in aging have shown positive effects on lifespan and "healthspan" (Childs et al., 2017; Gasek et al., 2021). However, senescent cell depletion experiments have yet to reveal specific mechanisms by which senescent cells influence cellular and molecular pathways that contribute to aging as well as chronic inflammatory processes.

Clues about how senescent cells alter local tissue environments have also been derived from their positive effects in tissue physiology (Muñoz-Espín et al., 2013; Ritschka et al., 2017). For example, transient accumulation of senescent cells modulates wound healing responses through IL-6 (a component of SASP; Mosteiro et al., 2016, 2018). Senescence regulates developmental pathways in embryogenesis (Muñoz-Espín et al., 2013; Storer et al., 2013; Yun et al., 2015), pregnancy, and its termination (Cox and Redman, 2017). Furthermore, it controls the finite

...................................................................................................................................................................

[1]Boehringer Ingelheim, Biberach an der Riß, Germany; [2]Max Planck Institute of Biochemistry, Martinsried, Germany; [3]Institute for Clinical Chemistry and Laboratory of Medicine, Faculty of Medicine at University Hospital, Technische Universität Dresden, Dresden, Germany; [4]Institute of Clinical and Experimental Trauma-Immunology, University Hospital Ulm, Ulm, Germany.

*D. Schloesser and L. Lindenthal contributed equally to this paper.   Correspondence to Karim Christian El Kasmi: karim_christian.el_kasmi@boehringer-ingelheim.com; Peter Murray: murray@biochem.mpg.de.

lifespan of red blood cells and their eventual destruction to recycle iron (Kay, 1975). Each of these positive effects of senescence involves stepwise elimination of unwanted cells in conditions of normal tissue homeostasis. Thus, cellular senescence is context dependent and temporally balanced; as aging proceeds or inflammation fails to resolve, this balance likely shifts towards the accumulation of senescent cells and associated detrimental disease-driving effects.

At present, little is known about the interplay between senescent cells and other cells. The senescence field has focused primarily on the effects of SASP as SASP components are readily accessible and quantifiable and contain constituents with known and predictable functions (e.g., IL-6). Conceivably, senescent cells could also modify the properties of neighboring cells through direct contact. As chronic diseases are not only associated with pro-inflammatory pathways but also with failure to restore pro-homeostatic processes (Nathan and Ding, 2010), we reasoned that senescent cells could influence neighboring non-senescent cells. Macrophages are the central immune cells required for tissue homeostasis and the cell type chiefly responsible for driving tissue-level inflammation, for example, in obesity, arthritis, chronic lung diseases and liver fibrosis, to mention a few (Eming et al., 2017; Murray and Wynn, 2011). Accordingly, we developed co-culture systems between primary human or murine macrophages and multiple types of senescent cells to dissect and define their interactions.

We found that senescent cells were refractory to macrophage-mediated engulfment and removal, consistent with the fact that senescent cells accrue in aging and chronic disease. Additionally, we discovered that senescent cells suppressed the ability of macrophages to remove apoptotic corpses. Senescent cell-mediated efferocytosis suppression (SCES) was common to all primary and transformed human and murine cells, including fibroblasts, liver stellate cells, and epithelial cells. SCES involved macrophage paralysis by augmented CD47 expression coincident with increased CD47-modifying enzymes QPCT/L in senescent cells. SCES was reversible by interfering with the SIRPα-CD47-SHP-1 axis or QPCT/L activity. While CD47 expression increased substantially in human and mouse senescent cells in vitro and in vivo, another ITIM-containing protein, CD24, contributed to SCES in human epithelial senescent cells and compensated for genetic deficiency in CD47. Thus, our findings implicate increased CD47 and CD24 expression as a component of the pathogenic effects of senescent cells mediating failure of homeostatic functions, such as efferocytosis, which must occur efficiently to maintain tissue homeostasis and suppress autoimmunity (Boada-Romero et al., 2020; Doran and Tabas, 2020; Morioka et al., 2019).

## Results

### Senescent cells are refractory to macrophage engulfment or killing

To quantify interactions between senescent cells and macrophages we generated senescent cell populations from primary and immortalized human and mouse cells. We used the CDK4/6 inhibitor palbociclib, the DNA damaging agent etoposide, γ-irradiation, or serial passaging to replicative senescence (Hayflick limit; Hayflick and Moorhead, 1961). These approaches yielded cell populations with hallmarks of senescence (Hernandez-Segura et al., 2018; Sharpless and Sherr, 2015), including β-galactosidase positivity (Fig. 1 A), increased cell size (Fig. 1 B), and increased CDKN1A mRNA encoding for the cell cycle inhibitor p21 (Fig. 1 C). As expected in all conditions of senescence induction, cellular proliferation was reduced, as shown by attenuated BrdU incorporation (Fig. 1 D). Predictably, protein and mRNA expression of senescence associated secretory phenotype (SASP) components (IL-6 and CXCL1; Fig. 1, E–H) were increased.

We used live cell imaging to monitor co-cultures between senescent fibroblasts from actin-GFP+ mice (generated by γ-irradiation, palbociclib, or Hayflick limit) and bone marrow-derived macrophages (BMDMs) from tdTomato+ mice seeded at different ratios (Fig. 2, A and B). Macrophages swarmed around and over large senescent cells but neither killed nor engulfed them. Quantification of the total green object area across a period of 48 h indicated no significant changes in the number of GFP+ senescent cells (Fig. 2 C). Importantly, we noted senescent cells were motile and constantly reoriented their membrane extensions, while macrophages were moving over and around them (Video 1). Similar findings were made with human senescent cells (both primary normal healthy lung fibroblasts [NHLF] and pancreatic epithelial cancer cells [Panc1]) co-cultured with primary human monocyte-derived macrophages (MDMs; Fig. 2 D). We also changed the activation state of the input primary murine and human macrophages by polarizing with either LPS or IL4 and IL13. Macrophage polarization to either extreme did not trigger senescent cell killing or engulfment by macrophages (Fig. S1, A and B). Therefore, in these two-cell in vitro co-cultures, senescent cells were neither killed nor engulfed by macrophages, an outcome consistent with the fact that living cells are refractory to macrophage phagocytosis.

### Senescent cells inhibit corpse removal by macrophages

Efferocytosis (corpse removal) is a core function of macrophages required continuously in all tissues to maintain homeostasis, suppress autoimmunity and chronic diseases, and promote resolution and restoration of homeostasis (Boada-Romero et al., 2020; Doran and Tabas, 2020; Gerlach et al., 2021; Morioka et al., 2019; Morioka et al., 2018; Yurdagul et al., 2020). We therefore tested if senescent cells modified the ability of macrophages to efferocytose corpses. Using a three-component co-culture system consisting of senescent cells, primary macrophages, and labeled cell corpses from UV-irradiated Jurkat or Raji cells, we employed independent approaches to control the timing of the interactions between each component in the culture system. First, human senescent or proliferating cells (the control in this system) and primary human MDMs were co-cultured and subsequently exposed to cell corpses (Raji cells). We quantified corpse engulfment using live cell imaging (Fig. 3 A), where we observed that senescent but not proliferating cells inhibited the ability of co-cultured MDMs to remove corpses. More specifically, senescent human primary NHLF, human fibroblasts derived from fibrotic lungs (IPF; Fig. 3 B), or human primary

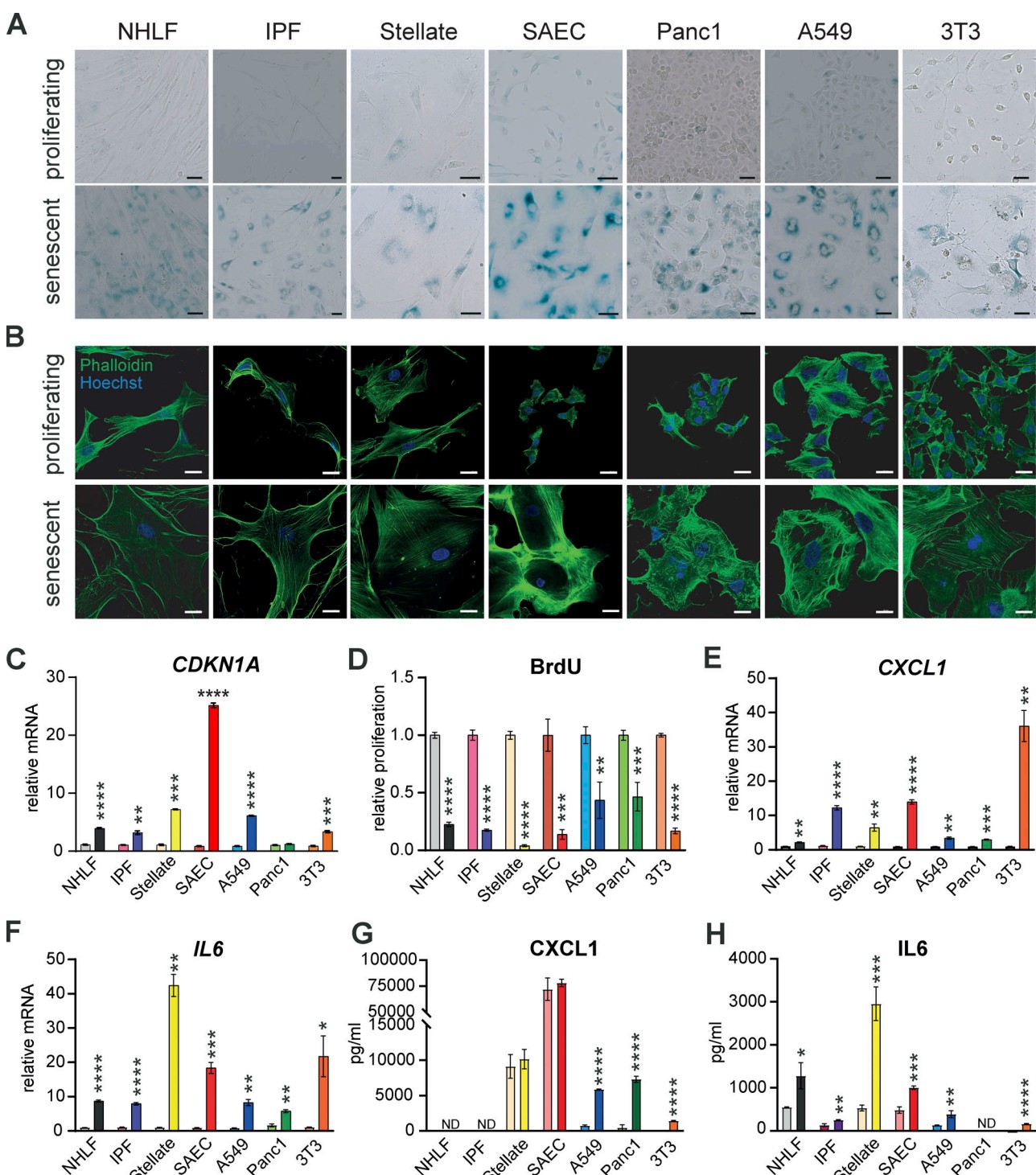

Figure 1. **Chemical treatment induces a senescent cellular phenotype.** Senescence was induced by palbociclib, except for A549 cells where etoposide was used. **(A)** Representative β-galactosidase staining was performed to mark senescent cells from indicated cell types after chemical induction of senescence. Scale bars indicate 50 µm. Images are representative of at least three independent experiments. **(B)** Representative immunofluorescence images from indicated cell types stained for Phalloidin (green) and Hoechst (blue). Scale bars indicate 20 µm. *n* = 2. **(C)** qPCR was performed to test *CDKN1A* gene expression in proliferating or senescent cells. Relative mRNA levels were normalized to the respective proliferating controls. Data are representative of three independent experiments. All values are means ± SEM. **P < 0.005, ***P < 0.0005, ****P < 0.0001. Statistically significant differences were determined by unpaired Student's *t* test. **(D)** Proliferation was quantified in the indicated proliferating or senescent cells using a BrdU Assay. Proliferation rate was normalized to the respective proliferating control. Data are representative of three independent experiments. All values are means ± SEM. **P < 0.005, ***P < 0.0005, ****P < 0.0001. Statistically significant differences were determined by unpaired Student's *t* test. **(E and F)** qPCR was performed to test *CXCL1* (E) and *IL6* (F) gene expression in proliferating or senescent cells. Relative mRNA levels were normalized to the respective proliferating controls. Data are representative of three independent experiments. All values are means ± SEM. *P < 0.05, **P < 0.005, ***P < 0.0005, ****P < 0.0001. Statistically significant differences were

determined by unpaired Student's *t* test. **(G and H)** CXCL1 (G) and IL6 (H) secretion of the indicated proliferating or senescent cells was quantified in the supernatant by ELISA. Data are representative of at least two independent experiments. All values are means ± SEM. *P < 0.05, **P < 0.005, ***P < 0.0005, ****P < 0.0001. Statistically significant differences were determined by unpaired Student's *t* test.

normal liver stellate cells (Fig. 3 C) all suppressed the ability of co-cultured MDMs to remove corpses relative to co-cultures with non-senescent, proliferating counterparts. Human senescent primary alveolar epithelial cells (SAEC) also reduced uptake of apoptotic corpses by co-cultured MDMs compared to co-cultures with proliferating cells (Fig. 3 C). We obtained similar results with co-cultures consisting of MDMs and senescent lung epithelial cancer cells (A549) or senescent pancreatic cancer cells (Panc1) compared to co-cultures with proliferating cells (Fig. 3 C).

As a complimentary approach to investigate apoptotic corpse removal and its degradation, we exposed macrophages co-cultured either to proliferating or senescent primary IPF-derived fibroblasts to apoptotic Raji co-labeled with a pH-insensitive dye (Cell tracker) and a pH-sensitive dye (CypHer5E). After fixation and subsequent immunofluorescent staining of CD45 and Hoechst, we analyzed efferocytosis using microscopy (Fig. 4 A). Macrophages co-cultured with senescent IPF-derived lung fibroblasts engulfed less apoptotic corpses compared to co-cultured with proliferating fibroblasts 1, 2 and 4 h post feeding (Fig. 4 B). Quantification of the overall green events after fixation indicated senescent fibroblasts (normal healthy and IPF-derived) reduced the uptake of green labeled apoptotic corpses by macrophages in comparison to proliferating fibroblasts (Fig. 4, C and D). We next calculated the ratio between cell tracker and CypHer5E double positive cells and the total number of cell tracker positive events to investigate the rate of acidification of the phagolysosome after apoptotic corpse engulfment (Fig. 4, E and F). In contrast to the engulfment rate, acidification of the phagolysosome did not differ between co-cultures of macrophages with primary lung fibroblasts (normal or IPF-derived).

We next explored the kinetics of corpse removal inhibition. Human MDMs were co-cultured with senescent A549 cells followed by exposure to unlabeled apoptotic corpses. 24 h after this first exposure to corpses, the same MDMs were re-exposed to the same amount of labeled apoptotic corpses to visualize corpse removal. In this scenario, removal of labeled corpses was also reduced compared to MDMs co-cultured with non-senescent proliferating cells (Fig. S1 C). These data established that suppression of corpse removal by senescent cells persisted over time. As a complimentary approach, we exposed murine BMDMs to murine senescent fibroblasts for either 6 or 24 h in co-culture, before adding apoptotic Jurkat cell corpses and measured efferocytosis by flow cytometry (1 h after exposure to corpses). Senescent mouse fibroblasts significantly suppressed removal of labeled apoptotic Jurkat cells by co-cultured primary mouse BMDMs relative to BMDMs in single culture at both time points, albeit more strongly at the 6 h time point (Fig. 5 A). This effect was independent of activation status as prior stimulation with LPS or IL4/IL13 did not influence the ability of macrophages to respond to local senescent cells in terms of efferocytosis inhibition (Fig. S1, D–F). To exclude the possibility that

human Jurkat cell corpses induced an unexpected xenobiotic reaction in murine co-cultures, we used primary murine BM-derived UV-irradiated eosinophil corpses as the target. As with Jurkat cells, senescent cells also blocked the ability of macrophages to engulf these corpses (Fig. 5 B). We next asked if the suppression of engulfment of cellular apoptotic corpses imposed by senescent cells on macrophages would extend towards suppression of phagocytosis of bacteria, dead cells, or inert beads. Senescent cells (including 3T3 cells, primary human healthy or IPF-derived lung fibroblasts, Panc1 and A549 cells) also inhibited engulfment of these particles by co-cultured macrophages relative to co-cultures with proliferating cells (Fig. S2). When viewed together, our findings suggest that senescent cells delivered a paralysis-type signal to macrophages that generally impeded efferocytosis and phagocytosis that we refer to hereafter as senescent cell-mediated efferocytosis suppression (SCES).

### SCES requires cell–cell contact

Senescent cells modify their local environment by SASP, which in principle could be a component of the mechanism through which senescent cells mediated phagocytosis suppression (Coppé et al., 2010). To test the role of SASP in SCES, we transferred conditioned media (CM) derived from 24 h cultures of senescent or proliferating cells (primary human fibroblasts or epithelial cells) to naïve human macrophages for 24 h followed by addition of apoptotic Raji corpses (Fig. 5 C). Under these conditions, corpse removal was not suppressed by CM derived from senescent cells when compared to CM derived from proliferating cells. As a second approach, we indirectly co-cultured senescent or proliferating cells with macrophages separated by a transwell (Fig. 5 D). The ability of macrophages to remove corpses was also not suppressed in indirect co-culture with senescent cells relative to indirect co-cultures with proliferating cells. Thus, these findings excluded an obvious role for SASP components in SCES and instead implicated cell–cell contact as a necessary step for SCES.

### CD47 increases in aging and disease

Efferocytosis is controlled by combinatorial "find me," "eat me" (e.g., surface phosphatidyl-serine), and "don't eat me" signals that ensure only dying or dead cells are removed (Lemke, 2019). "Eat me" and "don't eat me" signaling is regulated by macrophage cell surface molecules bearing intracellular immunoreceptor tyrosine-based inhibitory motif (ITIM) and immunoreceptor tyrosine-based activation motif (ITAM) motifs, which ultimately promote or suppress phagocytosis (Rumpret et al., 2020). The best understood macrophage ITIM "don't eat me" signal emanates from SIRPα. SIRPα binds to CD47 on neighboring cells, whose main function is to deliver a "don't eat me" signal and thereby suppresses cell engulfment. The potent "don't eat me" signal from CD47 gave rise to therapeutic approaches to disrupt this signaling

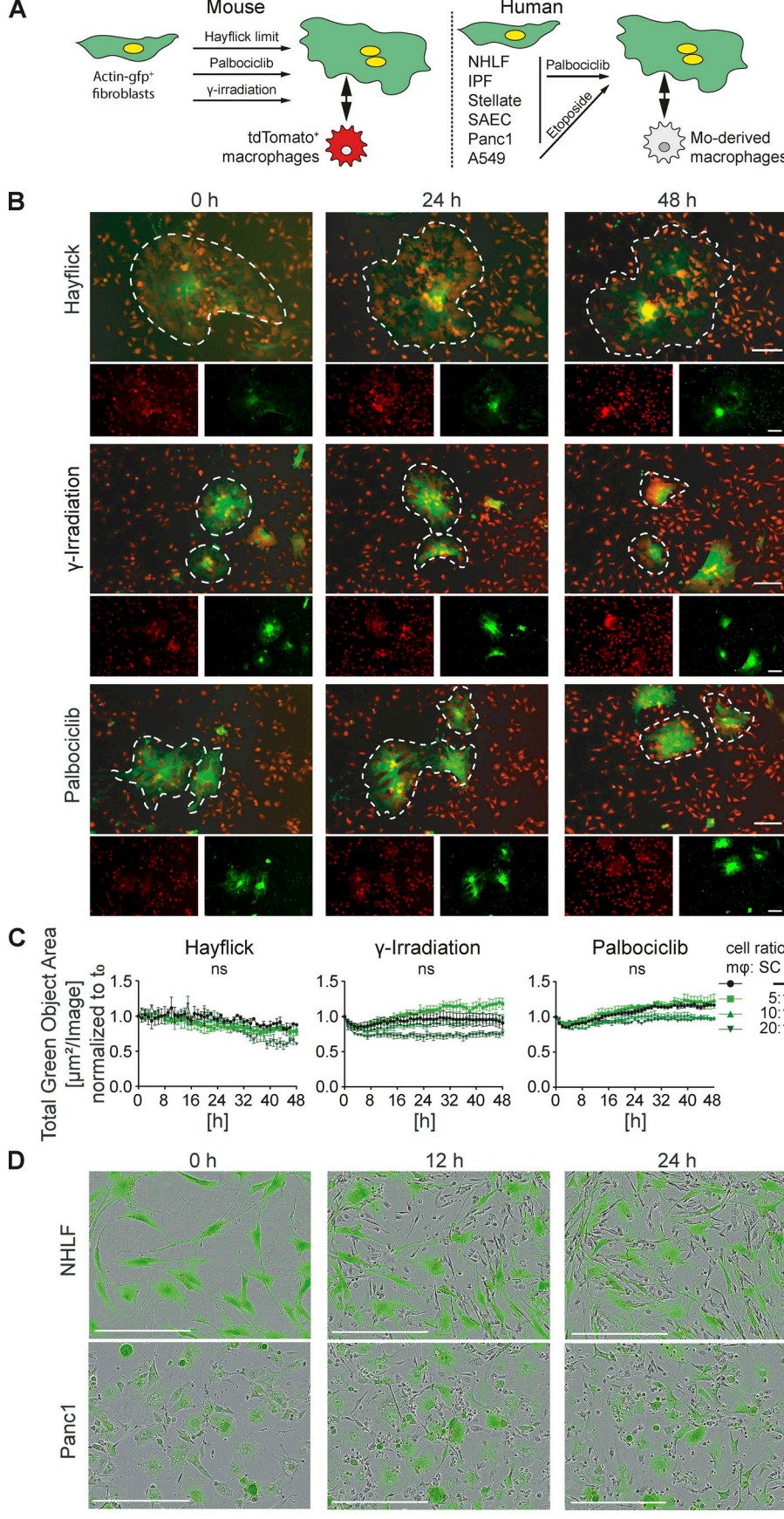

Figure 2. **Macrophages interact with but do not engulf or remove senescent cells.** **(A)** Schematic representation of senescence induction in murine and human primary and transformed cells. **(B)** Senescence was induced in actin-GFP mouse embryonic fibroblasts (MEFs) by either passaging stress until the Hayflick limit was reached by γ-irradiation or by treatment with palbociclib. Senescent MEFs were co-cultured with BMDMs, isolated from tdTomato⁺ mice, in a ratio of 10 BMDMs to 1 senescent cell and imaged over 48 h. In each image group, the same field of view was shown across time. Data are representative of three independent experiments; scale bars indicate 100 μm. **(C)** Quantification of green fluorescent area signal of senescent cells in co-culture with different ratios of BMDMs as indicated. Data are representative of three independent experiments. All values are means ± SEM. Statistically significant differences were determined by two-way ANOVA with Bonferroni correction. **(D)** Senescent NHLF (upper panel) or senescent human pancreatic epithelial cancer cells (Panc1; lower panel) stained with green Cytolight dye and co-cultured with human MDMs (unstained). Images were taken over 24 h. In each image group, the same field of view was shown across time. Data are representative of three independent experiments; scale bars indicate 200 μm.

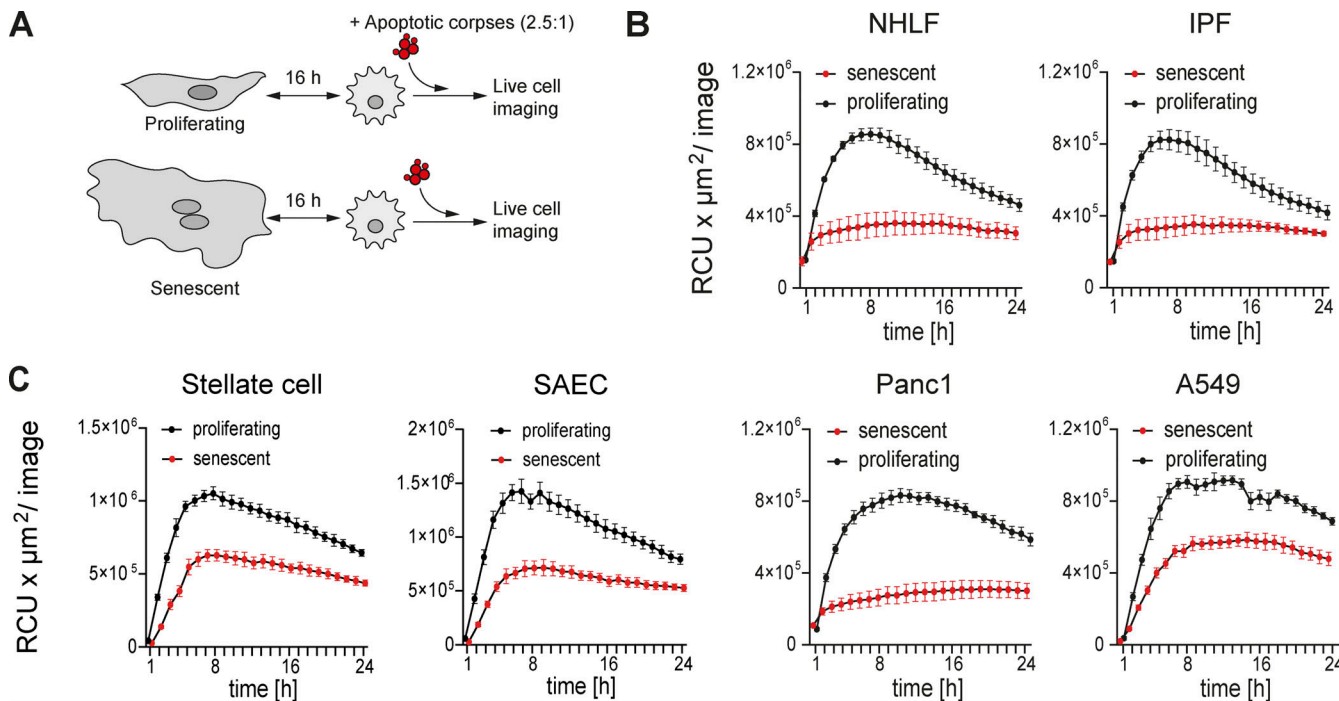

Figure 3. **Senescent cells impair macrophages' ability to remove corpses. (A)** Schematic overview of the experimental design using direct co-cultures between proliferating or senescent cells and primary macrophages. Cells were co-cultured for 16 h and then exposed to pHrodo labeled apoptotic corpses (Raji cells; 2.5:1 ratio). Corpse removal was monitored by live cell imaging (IncuCyte) for 24 h. **(B and C)** Quantification of efferocytosis of apoptotic corpses in co-cultures between human MDMs and senescent or proliferating primary lung fibroblasts (B; NHLF or IPF-derived fibroblasts [IPF]), primary liver stellate cells, primary lung small airway epithelial cells (SAEC), or transformed epithelial cell lines (C; Panc1, A549). Efferocytotic capability of macrophages in co-culture with proliferating (black line) or senescent (red line) cells was monitored over time using the IncuCyte S3 system. Data are representative of at least three independent experiments. All values are means ± SEM.

pathway, most prominently to enhance macrophage engulfment of malignant cells (Logtenberg et al., 2020). Accordingly, we reasoned that one or more "don't eat me" signals could have increased expression or activity on senescent cells relative to proliferating cells: thus, a corresponding CD47 increase should be observed in in vivo situations with increased numbers of senescent cells, such as normal aging, and chronic diseases of the lungs and liver. We interrogated the Tabula muris senis, a pan-tissue single cell sequencing atlas of murine aging from 1 to 30 mo of age (Tabula Muris, 2020) and found CD47 mRNA expression increased, parallel to increased senescence markers (Cdkn1a) in lung tissue with age. By contrast, no obvious changes were observed in SIRPα mRNA expression (Fig. 6 A). Similarly, re-analysis of the RNA-seq data set derived from bile duct ligation (BDL) or thioacetamide (TAA) induced rat liver fibrosis models revealed increased CD47 and SIRPα levels in fibrotic livers, corresponding to elevated Cdnk1a levels (Fig. 6, B and C; Wang et al., 2021). In subcutaneous adipose tissue (SAT) and visceral adipose tissue (VAT) of mice chronically fed a high fat diet, we found elevated CD47 expression compared to fat tissue derived from lean mice, which tracked with p21+ cells (Fig. 6, D and E) linking increased CD47 expression to a senescent cellular phenotype. Thus, various pathophysiological scenarios are linked to the accumulation of senescent cells and expression of CD47 (McHugh and Gil, 2018; Palmer et al., 2019; Schafer et al., 2017).

### Increased CD47 in murine and human senescent cells

We next tested if senescent cells in vitro also exhibited elevated CD47 expression. We noted a marked increase in CD47 mRNA expression relative to proliferating cells in primary human senescent cells and cell lines as well as in senescent murine fibroblasts (Fig. 7 A). These findings were corroborated by increased CD47 immunofluorescence (Fig. 7, B and D) and protein expression (Fig. 7, C and E). Notably, CD47 expression increased over time when primary lung fibroblasts were made senescent by serial passaging (Fig. 7 F) and this phenotype tracked with increased p21 (Fig. 7 G). Thus, augmented CD47 expression is a hallmark of cellular senescence in vitro and in vivo and could be the mechanistic link that mediates SCES.

### Fibroblast CD47 mediates SCES

To link increased CD47 expression to SCES, we generated a Cd47−/− murine 3T3 line and induced senescence in these cells with palbociclib. Efficient genome editing was confirmed by the complete absence of CD47 protein expression by flow cytometry (Fig. 8 A) and immunofluorescence (Fig. 8 B). Loss of surface CD47 did not result in clearance or engulfment of senescent cells by co-cultured macrophages (Fig. S3 A). Importantly, senescent 3T3 cells lacking CD47 failed to suppress corpse removal by co-cultured macrophages (Fig. 8 C). Blocking CD47 (Fig. 8 D) on human senescent cells (primary lung fibroblasts derived from

Figure 4. **Senescent cells reduce the uptake of apoptotic corpses by macrophages. (A)** Schematic overview of the experimental design using direct co-cultures between proliferating or senescent fibroblasts (normal healthy or IPF-derived) and primary macrophages. Cells were co-cultured for 16 h, and then

exposed to Cell tracker green and CypHer5E co-labeled apoptotic corpses (Raji cells; 2.5:1 ratio). Co-cultures were washed 1, 2 or 4 h post feeding and subsequently fixed to perform immunofluorescence staining. **(B)** Immunofluorescence microscopy of MDM in indicated co-cultures (either proliferating or senescent primary IPF-derived fibroblasts) exposed to apoptotic Rajis (labeled with Cell tracker green [green] and the pH-sensitive dye CypHer5E [white]) for 1, 2 or 4 h were then washed, fixed, and stained for CD45 (red) and Hoechst (blue). Scale bar: 20 μm. Representative images are from two independent experiments. **(C and D)** Quantification of apoptotic corpse uptake by macrophages in co-culture with human MDMs and senescent or proliferating primary IPF-derived (C) or normal healthy fibroblasts (D). Measurements and analysis of the number of green events per image (cell tracker green labeled Raji cells) were performed using the IncuCyte S3 system. All values are means ± SEM. Statistically significant differences were determined by two-way ANOVA with Tukey correction; n = 6 biological replicates. *P < 0.05, **P < 0.001. **(E and F)** Quantification of the ratio between red (CypHer5E)+ green (Cell tracker)+ (overlap) events and the total number of green events in the same field of view. Data are obtained from co-cultures between MDMs and proliferating (black) or senescent (red) IPF-derived (E) or normal healthy fibroblasts (F) exposed to apoptotic Rajis as shown in A. All values are means ± SEM. n = 6 biological replicates.

healthy [Fig. 8 E] or IPF-lungs [Fig. 8 F]) using anti-CD47 FAB fragments augmented corpse removal by co-cultured MDMs relative to co-cultures without CD47 blockade. Therefore, the CD47 pathway is essential for SCES.

## Senescent epithelial cells increase both CD24 and CD47 expression

So far, the majority of our mechanistic experiments focused on senescent fibroblasts as a model system. Conceivably, however,

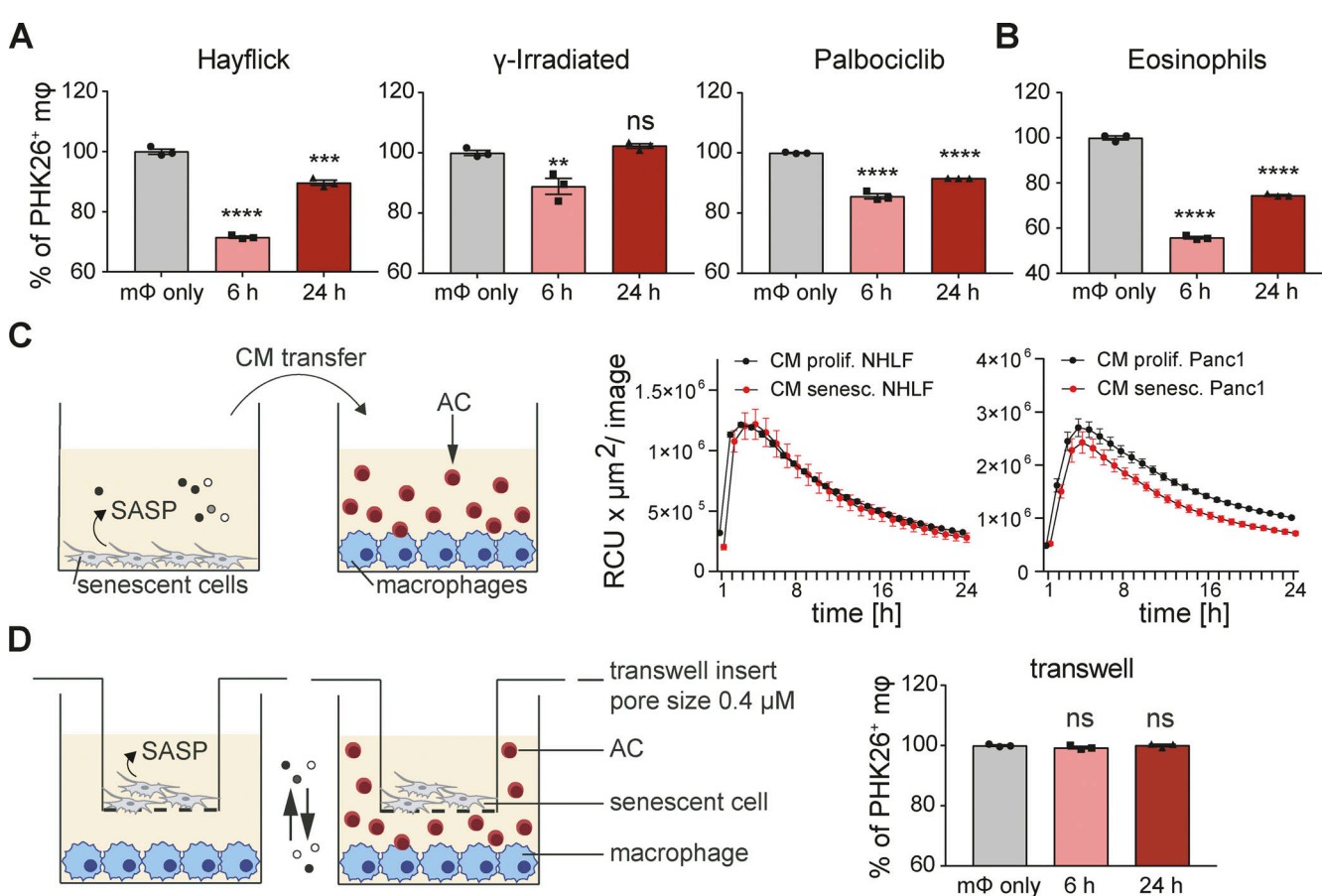

Figure 5. **Senescent cells inhibit corpses removal through direct cell-cell contact. (A)** Quantification of efferocytosis of apoptotic corpses by BMDMs in the presence of different types of murine 3T3 senescent cells (senescence induction by Hayflick limit, γ-irradiation or Palbociclib). BMDMs were primed in co-culture for 6 or 24 h. Apoptotic corpses (Jurkat cells) were labeled with PKH26 and added to macrophages in a 5:1 ratio. Samples were analyzed by flow cytometry 1 h post corpse exposure. All values in are means ± SEM; **P < 0.01, ***P < 0.001, ****P < 0.0001. Statistically significant differences were determined by one-way ANOVA with Bonferroni correction; n = 3 biological replicates. **(B)** Quantification of efferocytosis of apoptotic eosinophils by BMDMs in the presence of senescent 3T3 cells (induced by Palbociclib). All values in are means ± SEM; ****P < 0.0001. Statistically significant differences were determined by one-way ANOVA with Bonferroni correction; n = 3 biological replicates. **(C)** Conditioned media (CM) derived from either proliferating (black line) or senescent cells (red line) was transferred to naïve human MDMs. pHrodo labeled apoptotic corpses (AC, Raji cells) were added to MDMs and efferocytosis was monitored over time using the IncuCyte S3 System. Data are representative of three independent experiments. All values are means ± SEM. **(D)** Senescent cells (3T3, Palbociclib) were seeded in a transwell insert, which was placed into a well containing BMDMs. Efferocytosis of apoptotic corpses (Jurkat cells) by macrophages co-cultured with senescent cells in a transwell was analyzed by flow cytometry. Apoptotic corpses were labeled with PKH26 and added in a 5:1 ratio to the part of the well containing BMDMs. All values are means ± SEM. Statistically significant differences were determined by one-way ANOVA with Bonferroni correction; n = 3 biological replicates.

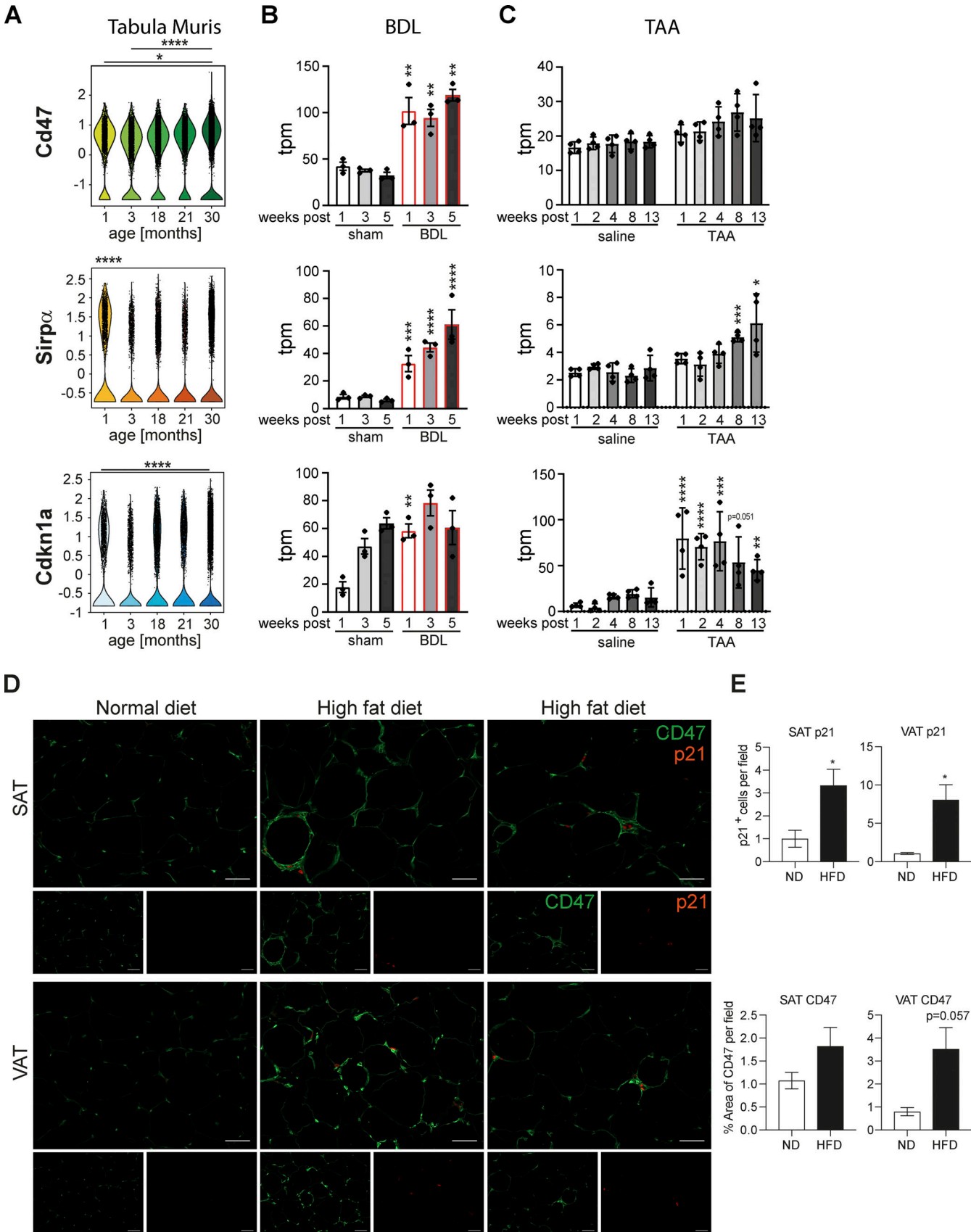

Figure 6. **Senescent cells exhibit increased CD47 expression in vivo and in vitro. (A)** Violin plots showing Cd47, SIRPα and Cdkn1a expression in lung tissue as assayed by the Tabula Muris Senis dataset. *P < 0.05, ****P < 0.0001. Statistically significant differences were determined by Wilcoxon test.

Expression levels obtained from RNA-sequencing data for liver fibrosis rat models (Wang et al. 2021). **(B and C)** Transcripts per million (tpm) expression values for *Cd47*, *Sirpα,* and *Cdkn1a* in rat livers derived from bile duct ligation (BDL; B) or thioacetamide (TAA)-induced liver fibrosis (C). Data are represented as mean ± SEM. Significant de-regulation (*P < 0.05, **P < 0.005, ***P < 0.0005, ****P < 0.0001) was determined by limma/voom based on counts derived from featureCounts. **(D)** Representative immunofluorescence images showing that p21 staining (red) co-localized with enhanced CD47 expression (green) in subcutaneous adipose tissue (SAT) and visceral adipose tissue (VAT) in mice upon feeding with a high fat diet. Scale bar: 50 μm. *n* = 3–4 mice per group. **(E)** Quantification of p21-positive cells per field and the percentual area of CD47 staining in SAT and VAT from animals fed a high fat diet (HFD) compared to normal diet (ND). All values are means ± SEM; *P < 0.05. Statistically significant differences were determined by *t* test or Mann-Whitney test, as appropriate; *n* = 3–4 mice per group.

any cell type can become senescent and emerging evidence suggests senescent epithelial cells have important contributions to the pathophysiology of chronic disease (Childs et al., 2017; Collado et al., 2007; He and Sharpless, 2017). Accordingly, we investigated the role of CD47 in epithelial SCES. As a model epithelial cell for genetic manipulation, we used pancreatic epithelial Panc1 cells to genetically ablate CD47, followed by senescence induction with palbociclib. CD47 expression was absent in gene-targeted senescent Panc1 cells (Fig. 8 G) and like *Cd47*⁻/⁻ 3T3 fibroblasts, *CD47*⁻/⁻ Panc1 cells were not engulfed by human macrophages (Fig. S3 B). Unlike results from the human and mouse fibroblasts systems, senescent *CD47*⁻/⁻ Panc1 mediated SCES (Fig. 8 H), suggesting the presence of additional "don't eat me" signals, either redundant to or overlapping with CD47 that could bypass the CD47-SIRPα pathway in SCES. CD24 is a GPI-anchored protein that interacts with the ITIM-containing inhibitory receptor Siglec-10 on macrophages and a newly recognized candidate to deliver "don't eat me" signals (Barkal et al., 2019). Immunoblot analysis revealed increased CD24 protein expression on senescent Panc1 cells relative to proliferating controls. In addition, both proliferating and senescent *CD47*⁻/⁻ Panc1 cells increased CD24 expression relative to WT Panc1 cells, suggesting that the lack of CD47 is in part compensated for by increasing CD24 (Fig. 8 G). Similarly, senescent epithelial A549 lung cancer cells and senescent primary human small airway epithelial cells (SAEC) also exhibited elevated CD24 expression relative to their proliferating counterparts (Fig. S3 C). Importantly, neither proliferating nor senescent fibroblasts expressed CD24 (Fig. S3, C and D), suggesting senescent epithelia use both CD47 and CD24 as "don't eat me" signals. Re-analysis of existing RNA-sequencing data sets for liver fibrosis rat models revealed increased CD24 (Fig. S3, E and F; Wang et al., 2021). Finally, neither epithelial cells nor fibroblasts expressed CD22 (Pluvinage et al., 2019), another surface molecule implicated in "don't eat me" processes (Fig. S3 C).

We next determined if CD24 on senescent epithelial cells contributed to SCES. Using anti-CD24 FAB fragments in co-cultures between senescent human epithelial *CD47*⁻/⁻ Panc1 cells and MDMs, we found that blocking CD24 in these co-cultures restored the ability of macrophages to remove corpses (Fig. 8 I). Based on these findings, we advanced the co-cultures to include senescent primary small airway epithelial cells (which express both CD24 and CD47) and MDMs and predicted that blockade of both CD47 and CD24 would reverse SCES. We used either CD47 or CD24 FAB fragments alone or in combination; as expected, only blocking both CD24 and CD47 increased corpse removal by co-cultured macrophages (Fig. 8 J), while CD24 or CD47 FAB fragments alone had no effect. Therefore,

both CD24 and CD47 are necessary and sufficient in senescent epithelial cells for SCES.

## SHP-1 mediates SCES

ITIM receptors like SIRPα or Siglec10 negatively regulate phagocytosis by different modes of signal strength and longevity (Rumpret et al., 2020). ITIM signaling recruits and activates SHP-1, a phosphatase, to dephosphorylate key proteins necessary for cytoskeletal rearrangement during target engulfment (Doran and Tabas, 2020; Lemke, 2019; Morioka et al., 2019). In case of SIRPα and Siglec10, activation and recruitment of SHP-1 in macrophages is likely to be involved in phagocytosis suppression. We visualized SHP-1 by immunofluorescence in human MDMs (visualized by CD45⁺ stain) co-cultured with senescent epithelial cells (Panc1; Fig. 9 A), which showed SHP-1 enrichment close to the macrophage plasma membrane. Similarly, we used murine tdTomato⁺ macrophages with senescent murine fibroblasts, followed by staining for SHP-1 (Fig. 9 B). When BMDMs were co-cultured for 6 h with *Cd47* wild-type senescent cells, SHP-1 staining was enriched beneath the macrophage plasma membrane. We quantified SHP-1 cellular location by measuring the relative enrichment across the diameter of the cell (Fig. 9, C and D). Validating those findings, SHP-1 recruitment to the membrane was reduced in BMDM co-cultures with *Cd47*⁻/⁻ senescent cells or BMDMs exposed to the broad SHP inhibitor (SHP1 and SHP2) NSC-87877 (Baker et al., 2011). SHP1 staining associated with the plasma membrane in macrophages co-cultured for 24 h with *Cd47* wild-type senescent cells was less prominent compared to macrophages co-cultured for only 6 h consistent with the data in Fig. 5 A showing that SCES is attenuated over time in the murine system. To functionally link SHP-1 to efferocytosis suppression in these experiments we inhibited SHP using NSC-87877 in mouse and human macrophage senescent cell co-cultures. In both cases, corpse removal by macrophages was augmented with NSC-87877 treatment compared to untreated co-cultures (Fig. 9, E–G). Therefore, SHP-1 signaling in macrophages contributes to SCES.

## QPCT/L activity is essential for CD47-mediated SCES

SIRPα binding to CD47 is dependent on N-terminal pyroglutamate modification catalyzed by QPCTL, a Golgi-resident enzyme, or its secreted family member QPCT (Logtenberg et al., 2020). The N-terminal pyroglutamate forms an important contact with SIRPα to stabilize the complex between macrophage and target cell (Hatherley et al., 2008). As CD47 on senescent cells is required for SCES, we reasoned a matched increase in QPCT/QPCTL expression and/or activity would ensure CD47 maturation in senescent cells in vivo and in vitro. Indeed, *Qpct*

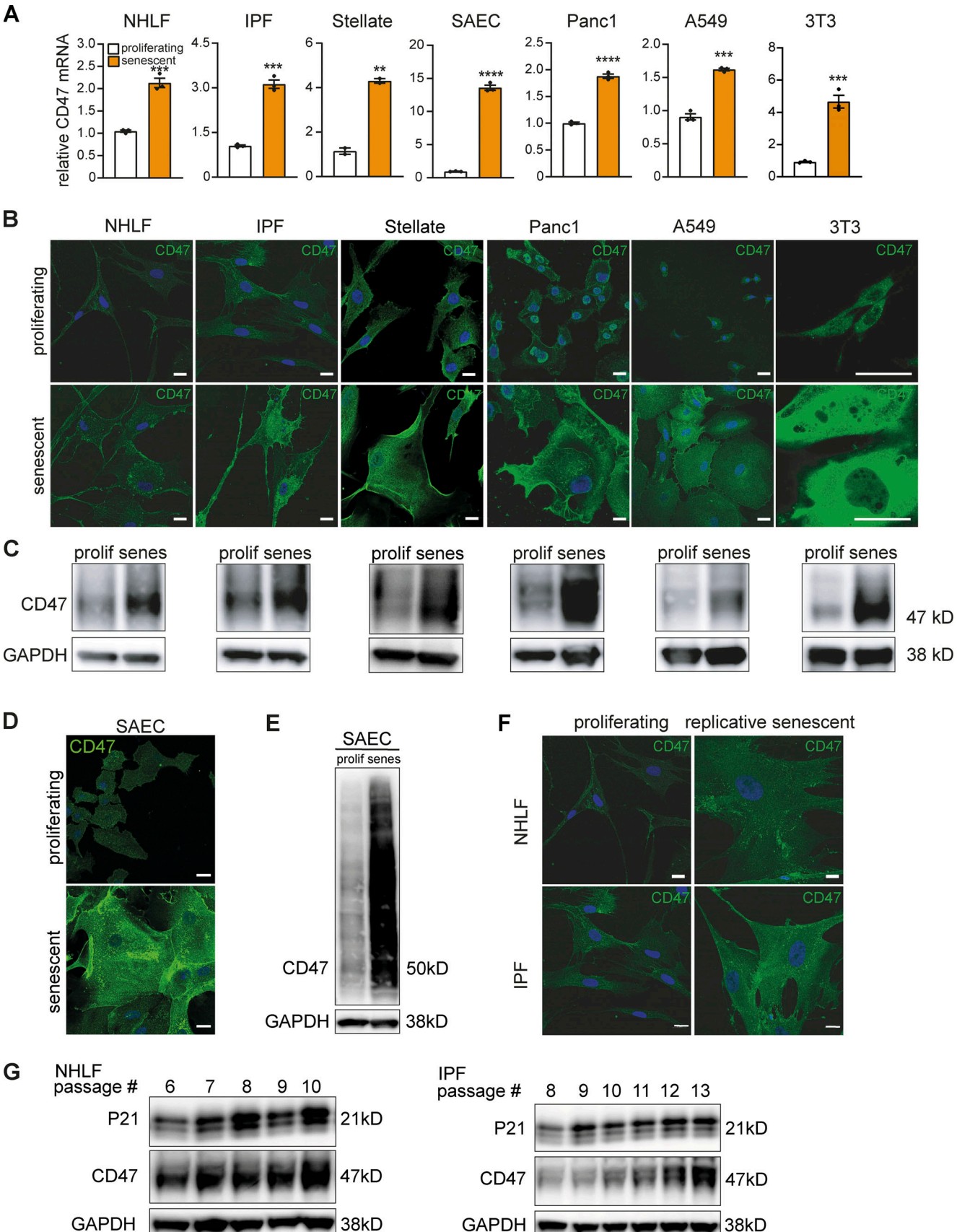

Figure 7. **Increased CD47 expression in aging and replicative senescence. (A)** Quantification of relative *CD47* mRNA levels by qPCR. Expression levels of senescent cells were normalized to the respective proliferating control (white bar). Data are representative of three independent experiments. All values are

means ± SEM. **P < 0.005, ***P < 0.0005, ****P < 0.0001. Statistically significant differences were determined by unpaired Student's *t* test. **(B)** Representative immunofluorescence images from indicated cell types stained for CD47 (green) and Hoechst (blue). Scale bar: 20 µm, except for 3T3 cells: 50 µm. **(C)** Whole-cell lysates from indicated cell types were isolated and analyzed by Western blotting for the indicated proteins. Senescence was induced by chemical treatment (palbociclib, except for A549: etopsoside). Shown are representative blots of three independent experiments. GAPDH images are derived from the same blot as Fig. S3 (for Panc1 cells the same GAPDH loading control is additionally used in Fig. 8 G). **(D)** Representative immunofluorescence images of proliferating or senescent small epithelial cells (SAEC) stained for CD47 (green) and Hoechst (blue). scale bars indicate 20 µm. *n* = 2–3. **(E)** Western blot analysis of CD47 expression in SAEC cells. Senescence was induced by chemical treatment (palbociclib). GAPDH was used as loading control. *n* = 2. GAPDH images are derived from the same blot as Fig. S3. **(F)** Representative immunofluorescence images of primary lung fibroblast (proliferative or replicative senescent by serially passaging; normal healthy [NHLF] or IPF-derived [IPF]) showing CD47 (green) and Hoechst (blue) staining. scale bars indicate 20 µm. *n* = 2. **(G)** Whole-cell lysates from primary NHLF or IPF lung fibroblasts were isolated and analyzed by Western blotting for the indicated proteins. Senescence was induced by replicative passing. Shown are representative blots of three to four independent experiments. Source data are available for this figure: SourceData F7.

and *Qpctl* mRNA expression in rat liver fibrosis models was elevated (Fig. S4, A and B; Wang et al., 2021). In addition, we found increased expression of *QPCT* and *QPCTL* mRNAs in murine and human senescent cells relative to proliferating cells (Fig. S4, C and D). Similarly, proteomic analyses of proliferating and senescent primary human lung fibroblasts (Fig. 10 A) or proliferating and senescent epithelial cells (Fig. 10 B) also revealed increased abundance of QPCTL and CD47 protein in senescent cells relative to proliferating cells. These changes in expression matched QPCT/L enzymatic activity (glutaminyl cyclase), which was increased in both murine and human senescent cells relative to proliferating cells (Fig. 10 C). Consistent with increased CD47 expression on senescent cells and increased amounts of its modifying enzyme QPCTL, binding of SIRPα was increased to senescent cells relative to proliferating cells (Fig. 10 D). Blockade of SIRPα using either anti-SIRPα blocking antibodies or FAB fragments (Fig. 10 E) partially reversed SCES (Fig. 10, F and G).

Currently, there are three classes of CD47-SIRPα signaling pathway inhibitors: blocking CD47 molecules, blocking SIRPα molecules, or small molecules that inhibit QPCT/L (Barreira da Silva et al., 2022; Logtenberg et al., 2020). To determine whether QPCT/L activity in senescent cells was required for SCES, we treated primary IPF-derived lung fibroblasts with the QPCTL inhibitor SEN177 (Jimenez-Sanchez et al., 2015) followed by senescence induction with palbociclib. We observed reduced QPCT/L enzyme activity in senescent primary IPF-derived fibroblasts (Fig. 10 H), which corresponded to reduced SIRPα binding to these senescent cells (Fig. 10 I). Importantly, upon glutaminyl cyclase inhibition with SEN177, senescent cells no longer suppressed corpse removal by co-cultured macrophages (Fig. 10 J and Fig. S4 E). Therefore, QPCT/L is an essential component required for CD47-mediated SCES.

## Discussion

We established that increased CD47 and CD24 expression on senescent cells negatively regulates macrophage-mediated corpse removal (efferocytosis), thereby interfering with a core homeostatic function of macrophages. We uncovered that senescent cells exert a potent "bystander" disease-driving mechanism through direct cell–cell contact, which was independent of the SASP and imposes a negative "don't eat it" signal onto macrophages. Additionally, we found that macrophages do not kill or engulf senescent cells; these findings are consistent with in vivo senescent cell accrual along aging and chronic inflammation.

Senescent cells are linked with multiple pathogenic sequelae in aging and disease (Childs et al., 2017; Collado et al., 2007; He and Sharpless, 2017; McHugh and Gil, 2018). Yet, how exactly senescent cells are pathogenic remains unclear. It is now well accepted that chronic diseases such as fibrosis are not only a consequence of activation of pro-inflammatory pathways but can also result from disruption of homeostatic tissue regulation and failure to engage pro-resolution pathways (Hernandez-Gonzalez et al., 2021; Nathan and Ding, 2010). Macrophages both promote disease and are critical for homeostasis and resolution (Eming et al., 2017; Murray and Wynn, 2011). For the latter functions, the ability to remove corpses (efferocytosis) is paramount and has been demonstrated to directly induce a pro-homeostatic and pro-resolving phenotype (Bosurgi et al., 2017; Gerlach et al., 2021; Morioka et al., 2019; Yurdagul et al., 2020). Thus, tissues containing large numbers of senescent cells could have an increased propensity to inhibit efferocytosis and phagocytosis, as in chronic tissue inflammation, fibrosis, and "inflammaging" (Nikolich-Žugich, 2018).

CD47 is ubiquitously expressed on normal cells, functioning as a "don't eat me" signal. Likewise, CD47 is overexpressed on many tumor cells (in relation to normal cells), allowing tumor cells to escape immune surveillance through inhibition of phagocytosis (Chao et al., 2012). This finding led to the concept that cancerous cells can "hijack" CD47 to escape immune surveillance, which subsequently triggered extensive therapeutic development towards disrupting the SIRPα-CD47 axis. The principal idea behind a therapy for the SIRPα-CD47 pathway is that an antibody or drug blocking SIRPα or CD47 would promote engulfment of a target coated in "don't eat me" signals. Our results expand on this concept and establish that senescent cells overexpress CD47 relative to normal proliferating cells and mediate SCES. In line with this, CD47 downregulation has been reported in chemotherapy-induced senescence escape (Guillon et al., 2019). Moreover, Wernig et al. (2017) demonstrated increased CD47 as a central process in fibrotic diseases and experimental CD47 blockade resulted in enhanced phagocytic activity together with attenuated fibrosis in vivo. In this regard, our observations coalesce with those in the cancer field, as CD47 expression may be an intrinsic property and hallmark of senescence, aging, and oncogenic transformation. However, our experiments using CD47-deficient murine or human senescent fibroblasts clearly show that loss of CD47 is not sufficient to trigger macrophage engulfment and other signals independent of CD47 restrict "self" phagocytosis. Translating these findings to

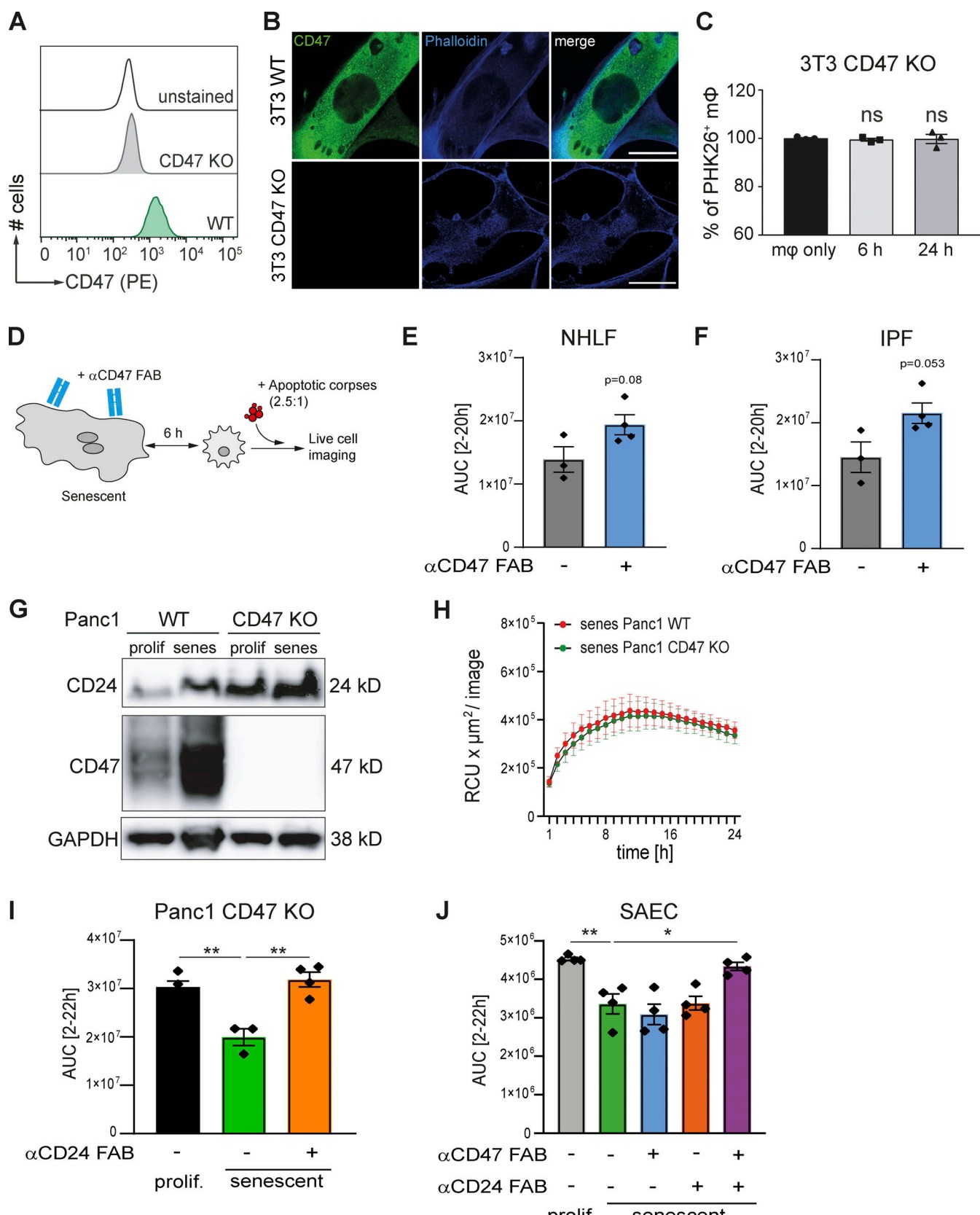

Figure 8. **Impairment of efferocytosis by senescent cells is mediated by CD47 and CD24 expression. (A)** CD47 loss-of-function in 3T3 cells was verified by CD47 staining in flow cytometry. **(B)** Representative immunofluorescence images from senescent 3T3 cells (WT or CD47 KO) showing CD47 (green) and Phalloidin (blue) staining (scale bar: 50 µm). *n* = 2. **(C)** Quantification of efferocytosis of apoptotic corpses by BMDMs in the presence of CD47 KO senescent

cells (induced by palbociclib). Samples were analyzed by flow cytometry. Data are representative of three independent experiments. All values are means ± SEM. Statistically significant differences were determined by one-way ANOVA with Bonferroni correction; n = 3 biological replicates. **(D)** Schematic overview of the experimental design. Senescent human fibroblasts were incubated with neutralizing anti-CD47 FAB fragments for 1 h, then direct co-cultures with primary macrophages were assembled for 6 h, followed by exposure to pHrodo labeled apoptotic corpses. Corpse removal was monitored by live cell imaging (In-cuCyte). **(E and F)** Quantification of efferocytosis of apoptotic corpses in co-cultures of human MDMs and senescent primary NHLF (E) or IPF-derived fibroblast (IPF; F). Efferocytotic capability of macrophages in co-culture with senescent fibroblast in the presence (blue bar) or absence of FAB fragments (gray bar) was monitored over time using the IncuCyte S3 system. Then area under curve (AUC) from 2 to 20 h was calculated and plotted. Data are representative of three independent experiments. All values are means ± SEM. Statistically significant differences were determined by unpaired Student's t test. **(G)** Whole-cell lysates derived from Panc1 cells (WT and CD47 KO) were analyzed by Western blotting for the indicated proteins. GAPDH was used as loading control. Senescence was induced by chemical treatment (Palbociclib). Blots are representative of three independent experiments. GAPDH image is derived from the same blot as Fig. 7 and Fig. S3. CD24 image is derived from the same blot as in Fig. S3. **(H)** Quantification of efferocytosis of apoptotic corpses by MDMs in the presence of senescent Panc1 cells. Efferocytotic capability of macrophages in co-culture with WT (red line) or CD47 KO (green line) Panc1 cells was monitored over time. Data are representative of at least three independent experiments. All values are means ± SEM. **(I)** Quantification of efferocytosis of apoptotic corpses in co-cultures of human MDMs and proliferating (black bar) or senescent CD47 KO Panc1 cells. Senescent CD47 KO cells were treated with (orange bar) or without anti-CD47 FAB fragments (green bar) prior to the assembly of the co-culture with MDMs. Efferocytotic capability of macrophages was monitored over time using the IncuCyte S3 system. Then area under curve (AUC) from 2 to 22 h was calculated and plotted. Shown values are means of at least technical triplicates ± SEM in one representative experiment. Data are representative of three independent experiments. **P < 0.01. Statistically significant differences were determined by one-way ANOVA with Bonferroni correction. **(J)** Quantification of efferocytosis of apoptotic corpses in co-cultures of human MDMs and proliferating (gray bar) or senescent primary small airway cells (SAEC). Senescent SEAC cells were either untreated (green bar) or treated with anti-CD47 (blue bar) or anti-CD24 FAB fragments (orange bar) or the combination of both FAB fragments (violet bar) prior to the assembly of the co-culture with MDMs. Efferocytotic capability of macrophages was monitored over time using the IncuCyte S3 system. Then area under curve (AUC) from 2 to 22 h was calculated and plotted. Shown values are means of technical quadruplicates ± SEM in one representative experiment. Data are representative of two independent experiments *P < 0.05, **P < 0.005. Statistically significant differences were determined by one-way ANOVA with Bonferroni correction. Source data are available for this figure: SourceData F8.

in vivo settings remains a challenge as ideally, one would need to determine the identity of senescent cells in tissues, and what combinations of "don't eat me" signals they express. In this regard, new tools to identify senescent cells in vivo through expression of cell cycle inhibitors (Reyes et al., 2022) could be used to ask if p16+ cells have increased relative CD47 and CD24 expression through aging, and the eventual fate of these cells.

More recent exploration of the cancer engulfment hypothesis has uncovered additional pathways including calreticulin, CD22, and CD24 that intersect with or run parallel to the SIRPα-CD47 pathway and are also potential targets in cancer (Barkal et al., 2019; Kamber et al., 2021; Pluvinage et al., 2019). Our data demonstrate that in comparison to senescent fibroblasts, senescent epithelial cells exploit both increased CD47 and increased CD24 expression, thereby delivering SHP1-dependent inhibitory signals that interfere with the efferocytotic capability of macrophages. Importantly, our data not only show that increased CD24 is an epithelial-specific mechanism for SCES but also indicate that increasing CD24 expression can compensate for loss of CD47, at least in our in vitro system. This result might be a clue for the limited success observed with anti-CD47 or anti-SIRPα centered therapies and might have important implications for the design and development of therapies that target the CD47-SIRPα axis. A key question for future exploration concerns the signaling pathways that control the increased CD47, CD24, and QPCT/L expression on senescent cells. While the SASP is not required for SCES, SASP components such as inflammatory cytokines could regulate the "don't eat me" machinery expressed on senescent cells. In theory, these signals could be uncovered by whole genome loss-of-function approaches.

Even though many have claimed that macrophages are critical for the removal of senescent cells, the accrual of senescent cells in aging and in disease is at odds with this notion. Further,

our experiments clearly indicate that the presence of macrophages (independent of their activation state) is insufficient to trigger engulfment of senescent cells. We argue that senescent cell removal in healthy tissues, normal aging, or disease likely involves other signals such as NK cell recognition of "disturbed self" (Ruscetti et al., 2018; Sagiv et al., 2013). However, Karin et al. have argued that the turn-over (and removal) of senescent cells is tied to aging itself: that is, senescent cell elimination slows as organisms age, accounting for the age-related accrual in senescent cells (Karin et al., 2019). In this case, senescent cell removal is semi-stochastic in nature and does not necessarily involve active surveillance pathways. Further work applying new senescent cell reporter animals in different immune and pathophysiological contexts could provide new insights into the dynamics of senescent cell numbers. A further variable in the interaction between senescent cells and macrophages involves target size evaluation by macrophages: senescent cells increase in size in vitro and in tissues, which may represent a natural barrier to phagocytosis (Reyes et al., 2022). Indeed, macrophage engulfment of tumor cells or senescent red blood cells (RBCs) may be facilitated by their small size. Further experiments to manipulate the size of macrophages versus their target proportions will be necessary to tie this phenomenon to, for example, macrophage polykaryon formation to attempt engulfment of large targets such as medical devices or asbestos fibers (Helming and Gordon, 2009; Milde et al., 2015).

## Limitations of the study
Modulation of the CD47-SIRPα-QPCT/QPCTL pathway in vivo requires experimental control over three variables: (i) a tissue environment that contains a predictable and quantifiable number of senescent cells, (ii) an assay to measure local macrophage-mediated efferocytosis, and (iii) predictable means of interrupting the CD47, SIRPα, or QPCT/QPCTL axis. At present, each

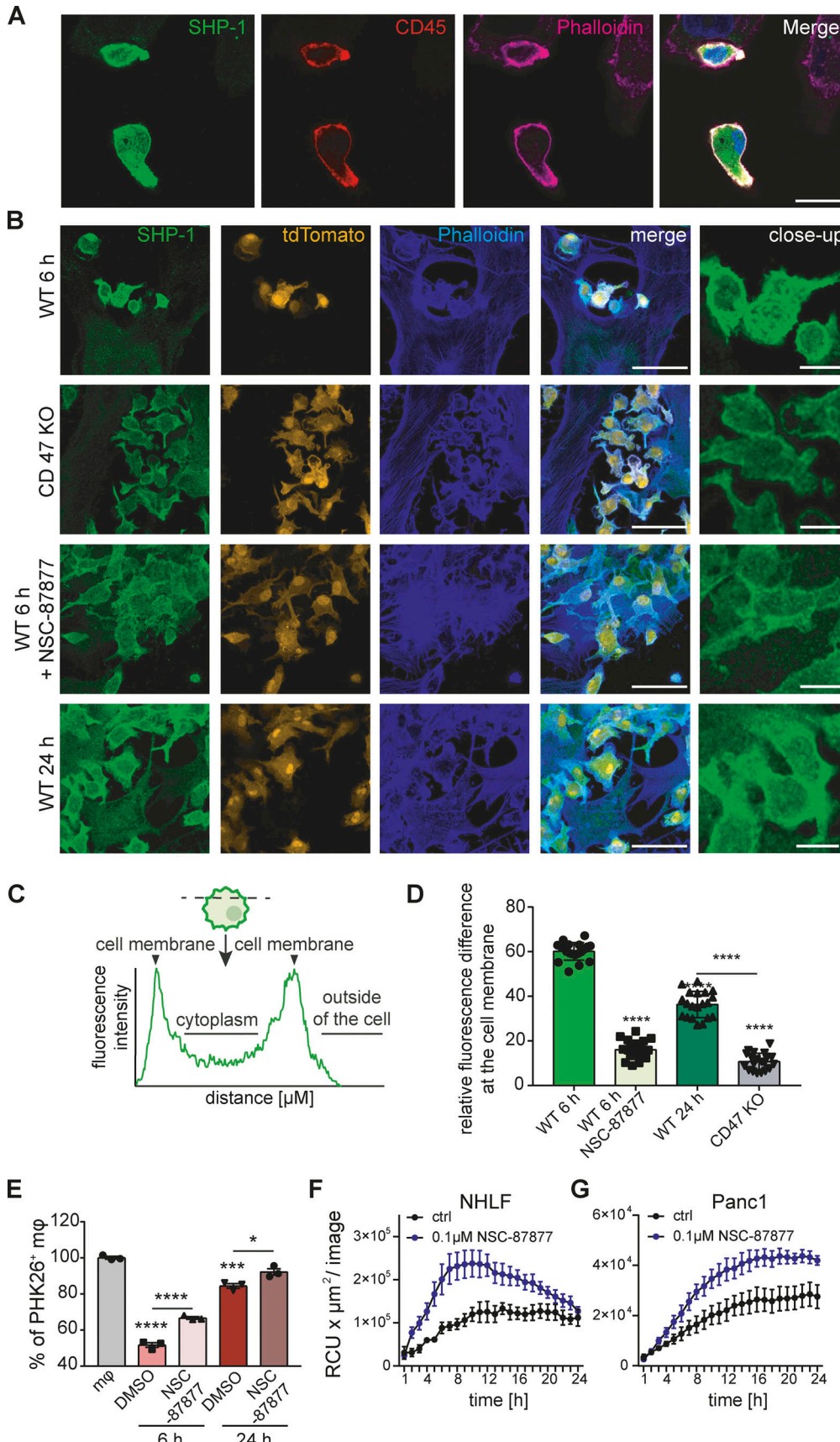

Figure 9. **SHP-1 mediates efferocytosis paralysis. (A)** Representative immunofluorescence images from direct co-cultures of senescent Panc1 cells and human MDMs. Cells were specifically stained for SHP-1 (green), macrophage marker CD45 (red), Phalloidin (magenta), and Hoechst (blue); scale bar: 20 μm.

*n* = 3. **(B)** BMDMs from tdTomato⁺(orange) mice were co-incubated with senescent WT or *Cd47*-deficient 3T3 cells for the times shown on the left of the images. Fixed samples were stained for SHP-1 (green) and Phalloidin (blue). Scale bars indicate 50 μm (main) or 12.5μm (inset). *n* = 2. **(C)** Schematic overview of the intensity profile measurement of SHP-1 in BMDMs. Intensity was measured across the widest point of each cell. Intensity profiles were generated by measuring the distance in μm and the gray value intensity. **(D)** Quantification of intensity profile measurement of SHP-1 in BMDMs. Data are representative of three independent experiments. All values are means ± SEM. Statistically significant differences were determined by one-way ANOVA with Bonferroni correction; ****$P < 0.0001$. *n* = 20 biological replicates. **(E)** Quantification of PHK26-labelled corpse efferocytosis in the presence of absence of NSC-87877 (SHP-1 and SHP-1 inhibitor). Data are representative of three independent experiments. All values are means ± SEM. Statistically significant differences were determined by one-way ANOVA with Bonferroni correction; *$P < 0.05$, ***$P < 0.001$, ****$P < 0.0001$; *n* = 3 biological replicates. **(F and G)** Quantification of efferocytosis of apoptotic corpses by MDMs in the presence of senescent NHLF (F) or Panc1 cells (G). Macrophages were pre-incubated for 1 h with NSC-87877, and then co-cultures were assembled prior to exposure to pHrodo labeled apoptotic corpses. Efferocytotic capability of macrophages with (blue line) or without inhibitor (black line) was monitored over time. Data are representative of three independent experiments. All values are means ± SEM.

of these variables can be addressed on their own, as aging and disease models replete with senescent cells are available (Kõks et al., 2016), complex models of in vivo efferocytosis exist (Roberts et al., 2017) and inhibitors of CD47 and QPCT/QPCTL have been developed and advanced for clinical use. The central challenge will be to combine these three components into a reproducible and timely experimental platform. Similar challenges exist for research into the different ways SASP regulates local tissue environments, or the ways in which small numbers of senescent cells induce wound healing processes (Mosteiro et al., 2016). Nevertheless, our data establish that CD47-SIRPα-QPCT/QPCTL is a component of senescent cell biology that may exert profound effects in vivo through the suppression of macrophage efferocytosis; most likely, efferocytosis inhibition is proportional to the number of senescent cells in tissues, arguing in our thinking, that the longer the life span or more severe the disease, efferocytotic corpses would accrue and be linked to inflammation perpetuation and/or other disease consequences.

In conclusion, we demonstrate that CD24 and CD47 deliver a hitherto undiscovered SHP1-dependent inhibitory senescent cell-derived signal that suppresses corpse removal and thus can serve as a platform for the development of therapeutics targeting the interaction of senescent cells and macrophages to restore pro-homeostatic functions.

# Materials and methods

## Mice

Actin-GFP mice constitutively expressing GFP under the control of the chicken beta-actin promoter were obtained from the Jackson Laboratories. C57BL/6-Tg(CAG-EGFP)1Osb/J. tdTomato⁺ mice were a gift of Dr. Christian Meyer (Max Planck Institute of Neurobiology, Martinsried, Germany) and were crossed to Tie2-Cre mice (B6.Cg-Tg(Tek-cre)1Ywa/J), which express Cre in all hematopoietic cells, permitting the isolation of tdTomato⁺ macrophages. Animal experimentation was performed at the Max Planck Institute of Biochemistry animal facility in accordance with and approval from the legal authorities ("Regierung von Oberbayern").

## Adipose tissue immunohistochemistry from lean and obese mice

Wild-type mice (C57BL6) were fed a control diet (D12450B, Research Diets, with 10% kcal fat) or a high fat diet (D12492, Research Diets, 60% kcal fat) for 20 wk. Experiments were

approved by the "Landesdirektion Sachsen." Subcutaneous or visceral adipose tissue was isolated from mice and embedded in paraffin after fixation with 4% PFA. For immunofluorescence staining for CD47 and p21, tissue sections were deparaffinized and antigen retrieval was done by antigen unmasking solution (H-3301-250; Tris-Based; Vector Laboratories). Sections were permeabilized with 0.1% Triton X-100 and blocked by serum-free protein block (X090930-2; Dako) and then incubated with primary antibody against p21 (ab188224; Abcam) and CD47 (AF1866; R&D) overnight at 4°C. After washing with PBS, sections were incubated with secondary antibody Donkey anti-Rabbit (H+L) Alexa Fluor 555 (A-31572; Invitrogen) and Donkey anti-goat (H+L) Alexa Fluor 488 (A-11055; Invitrogen) for 90 min at RT and counterstained with DAPI (D9542; Sigma-Aldrich). To reduce tissue autofluorescence, sections were incubated with TrueBlack Lipofuscin Autofluorescence Quencher (#23007; Biotium) for 30 s and mounted. Images were acquired using a ZEISS Axio Observer. Z1 with Apotome II. For the quantification of images, DAPI and p21 double-positive cells per field were counted; moreover, CD47 positive area per field was quantified from 4 to 5 images per mouse using Fiji software.

## Cell lines

All cells were purchased from the American Type Culture Collection (ATCC), except for primary healthy lung fibroblasts (NHLF) and lung fibroblasts derived from IPF lungs, which were obtained from Lonza. Raji cells and Jurkat cells were cultured in RPMI 1640 (61870-10; Gibco), supplemented with 10% fetal bovine serum (10500-064; FBS; Gibco), and 100 U/ml penicillin/streptomycin (15140-122; Gibco). A549 cells and Panc1 cells were cultured in DMEM (31966-021; Gibco), supplemented with 10% FBS and 100 U/ml penicillin/streptomycin (15140-122; Gibco). Primary lung fibroblasts were cultured in DMEM, supplemented with 10% FBS, 100 U/ml penicillin/streptomycin and non-essential amino acids (11140-035; Gibco). Primary human liver stellate cells were cultured in SteCM Medium (5301; ScienCell), while primary human small airway epithelial cells were cultured in PneumaCult-Ex Plus Medium (05040; StemCell). All cells were cultured at 37°C in a humified incubator with 5% $CO_2$. All cell lines were regularly tested by PCR and verified to be free of mycoplasma contamination.

## Isolation of primary mouse embryonic fibroblasts

Primary mouse embryonic fibroblasts (MEFs) were collected from sacrificed pregnant (E14-16) actin GFP mice. Intact uteri

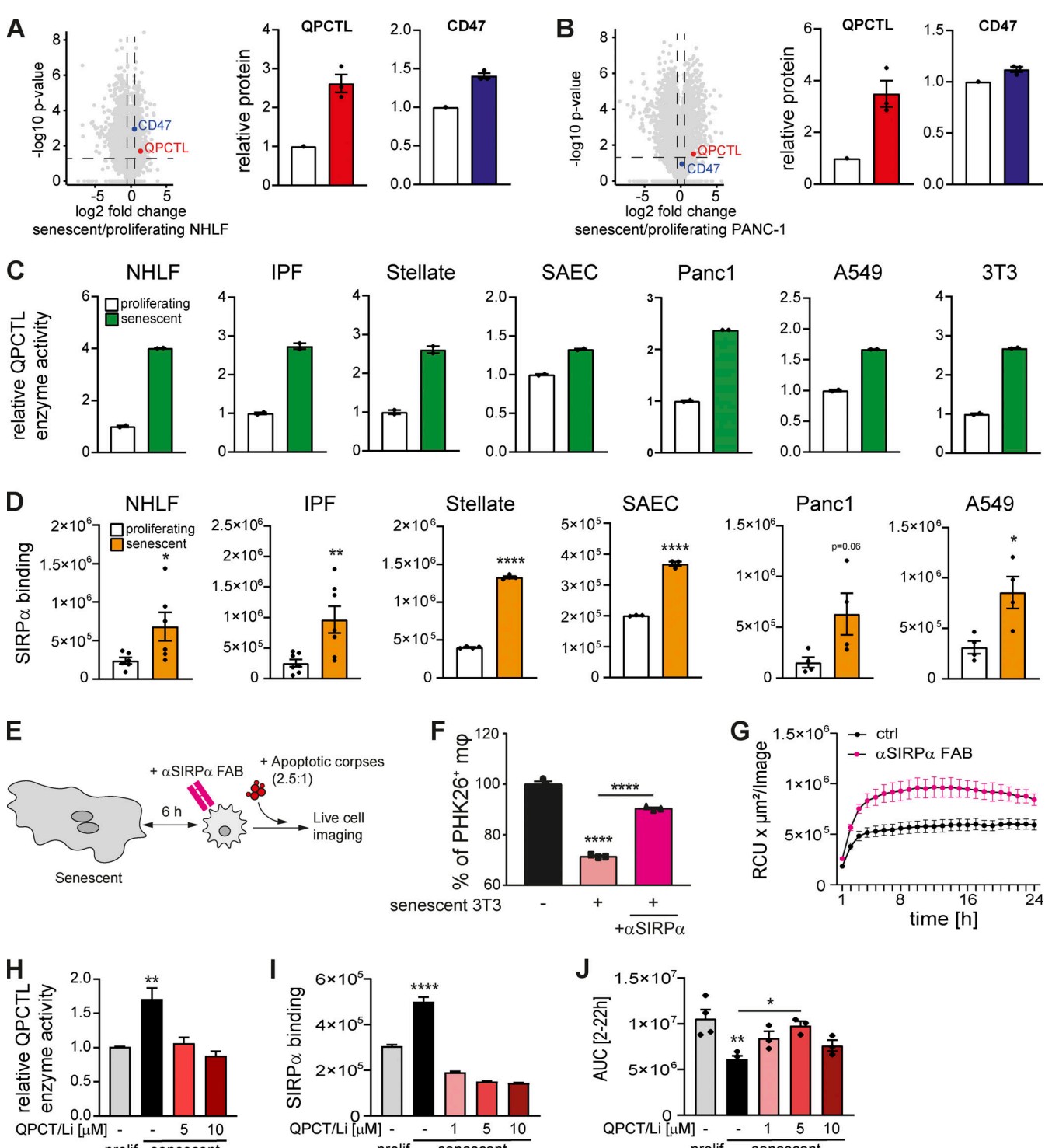

Figure 10. **QPCT/L is essential for mediating the suppression of efferocytosis through the CD47-SIRPα axis. (A and B)** Proteomic analysis of proliferating versus senescent NHLF (A) and Panc1 cells (B). Volcano plot representing the negative log10-transformed P values vs. the log$_2$ fold changes in protein intensities of senescent compared to proliferating cells. Dotted lines depict a P value of 0.05 and a fold change of 1.5. Senescence was induced by chemical treatment. Relative QPCTL (red symbol) and CD47 (blue symbol) protein expression normalized to the proliferating control. $n = 3$ biological replicates. **(C)** Quantification the relative QPCTL enzyme activity. Enzyme activity of senescent cells was normalized to the respective proliferating control (white bar). Data are representative of two independent experiments. All values are means ± SEM. **(D)** SIRPα binding of proliferating (white bar) or senescent cells (orange bar) was quantified using a luminescent reporter assay. All values are means ± SEM. Statistically significant differences were determined by unpaired Student's $t$ test; $n = 3$–6 biological replicates. **(E)** Schematic overview of the overall experimental design. Senescent fibroblasts were incubated with neutralizing anti-SIRPα antibodies (mouse) or FAB fragments (human) for 1 h, then direct co-cultures with primary macrophages were assembled for 6 h, and then exposed to labeled apoptotic corpses. Corpse removal was monitored by flow cytometry (mouse) or live cell imaging (human). **(F)** Efferocytosis in the presence of senescent 3T3 cells (induced by palbociclib) and a SIRPα blocking antibody. Data are representative of three independent experiments. All values are means ±

SEM. Statistically significant differences were determined by one-way ANOVA with Bonferroni correction; ****$P < 0.0001$. $n = 3$ biological replicates. **(G)** Quantification of efferocytosis of apoptotic corpses in co-cultures of human MDMs and senescent primary IPF-derived fibroblast (IPF). Efferocytotic capability of macrophages in co-culture with senescent fibroblast in the presence (magenta line) or absence of FAB fragments (black line) was monitored over time using the IncuCyte S3 system. Data are representative of three independent experiments. All values are means ± SEM. **(H)** QPCTL activity was quantified in proliferating or senescent primary IPF-derived lung fibroblasts in the presence (5, 10 μM) or absence of the QPCTL inhibitor SEN-177. Enzyme activity of senescent cells was normalized to the respective proliferating control (gray bar). Data are representative of four independent experiments. All values are means ± SEM. **$P < 0.005$. Statistically significant differences were determined by one-way ANOVA with Tukey correction. **(I)** SIRPα binding of proliferating or senescent primary IPF-derived lung fibroblasts in the presence (1, 5, 10 μM) or absence of the QPCTL inhibitor SEN-177 was quantified using a luminescent reporter assay. Data are representative of three independent experiments. All values are means ± SEM. ****$P < 0.0001$. Statistically significant differences were determined by one-way ANOVA with Bonferroni correction. **(J)** Quantification of efferocytosis of apoptotic corpses in co-cultures of human MDMs and proliferating (gray bar) or senescent IPF-derived lung fibroblasts. Senescent IPF-derived lung fibroblasts were treated with the QPCTL inhibitor SEN-177 (1, 5, 10 μM) prior the assembly of the co-culture with MDMs. Efferocytotic capability of macrophages was monitored over time using the IncuCyte S3 system. Then area under curve (AUC) from 2 to 22 h was calculated and plotted. Data are representative of three independent experiments. All values are means ± SEM. *$P < 0.05$, **$P < 0.005$. Statistically significant differences were determined by one-way ANOVA with Bonferroni correction.

were removed, washed with cold PBS, and each embryo individually prepared by removing appendages including the head and internal organs. A surgical scalpel blade was used to finely mince the remaining tissue in 100 μl of DNAse (1 mg/ml in $H_2O$) and 2 ml of 0.25% trypsin. The suspension was then applied through a 5 ml serological pipette. Suspended cells derived from each embryo were cultured in DMEM composed of 10% FBS, 1% penicillin/streptomycin. Cells were further cultivated and cryopreserved in freezing medium at passage 4 (P4) providing a residual stock for replicative senescence.

### Senescence induction
*Replicative senescence*: MEFs were cultured to passage 5–8 until further cell division ceased and cells manifested a size increase and β-galactosidase positivity. Human primary lung fibroblasts were serially passaged to induce natural senescence. *Palbociclib*: Cells were plated (96-well plate: $4 × 10^3$ cells/well) and subsequently rested over night before exposure to 4.5 μM Palbociclib. Palbociclib exposed cells were subsequently cultured for 6 d (without media change). *γ-Irradiation (Murine)*: MEFs were cultured to 90% confluence in T175 flasks. Two flasks containing the same embryo were harvested with 2 ml of 0.25% trypsin and pooled in one 50 ml conical polypropylene tube. The tube was subjected to 10 Gy γ-irradiation using a calibrated Cesium-137 source (Gamma Cell 40, Atomic energy of Canada Limited). The irradiated cells were directly cryopreserved in freezing medium without further passaging.

### Primary macrophage cell culture
*Murine BMDMs:* Mouse bone marrow was obtained by flushing femurs and tibiae with PBS followed by differentiation into BMDMs by culturing in complete Dulbecco's modified Eagle's medium (DMEM; 41966-029; Thermo Fisher Scientific) composed of 10% fetal bovine serum (S0115; Biochrome), penicillin-streptomycin, and recombinant human CSF1 (50 ng/ml) for 7 d as previously described (Dichtl et al., 2022). *Human MDMs*: All experiments with blood from human volunteer donors were approved by the blood donation service from Boehringer Ingelheim Pharma GmbH and Co. KG following ethical standards and local regulation. PBMCs of healthy donors were isolated from whole blood using density centrifugation (227290; Greiner; Fikoll gradient; Leucosep separation tubes). After washing with PBS three times, human CD14+ monocytes were isolated from

PBMCs using magnet beads (Pan-Macrophage isolation Kit) and using auto-macs system according to manufactures instructions (130-096-537; Miltenyi). Monocytes were plated in T175 cell culture flasks (18–20 million cells per flask) and differentiated into macrophages for 7 d using complete DMEM (31966-021; Gibco), supplemented with 10% FBS (Gibco; 10500-064), 100 U/ml penicillin/streptomycin (15140-122; Gibco) and non-essential amino acids (11140-035; Gibco) in the presence of M-CSF (50 ng/ml; 130-096-493; Miltenyi). After differentiation, macrophages were collected by gently scraping (83.1831; Sarstedt) and seeded in complete DMEM in the presence of 20 ng/ml M-CSF. All cell culture procedures were carried out at 37°C with 5% $CO_2$.

### Macrophage activation
MDMs and BMDMs were differentiated, detached, and seeded in 96-well as described above. Where indicated, BMDMs were stimulated with 10 ng/ml IL4 (404-ML-010/CF; R&D), 10 ng/ml IL13 (413-ML/CF; R&D), or 100 ng/ml LPS from *Escherichia coli* O55:B5 (L2880; Sigma-Aldrich) 1.5 h after the assembly of the co-culture with senescent cells. MDMs were stimulated with 10 ng/ml IL4 (204-IL-010/CF; R&D), 10 ng/ml IL13 (213-ILB-025/CF; R&D), or 100 ng/ml LPS from *Salmonella enterica* serotype typhimurium (L6143; Sigma-Aldrich) 1.5 h after the assembly of the co-culture with senescent cells.

### Isolation of bone marrow derived eosinophils
Bone marrow cells were placed in eosinophil medium containing 20% fetal bovine serum, 55 μM β-mercaptoethanol, 10 mM nonessential amino acids, 1 mM sodium pyruvate, 1% penicillin/streptomycin, 2 mM L-glutamine, and 25 mM N-2-hydroxy-ethylpiperazine-N′-2-ethanesulfonic acid (HEPES) buffer (Biochrome). Cells were stimulated with 100 ng/ml of stem cell factor (SCF) and 100 ng/ml of Fms-like tyrosine kinase 3 ligand (FLT3L) for 4 d at a density of $1 × 10^6$ cells per ml in a T75 flask. From day 4 on, cells were kept in medium supplemented with 10 ng/ml of recombinant murine IL-5. At days 6, 8, 10 and 12, half of the culture medium was replaced by fresh medium. At day 8 the whole culture was subjected into a new flask. From day 10, cell density was monitored to keep the concentration at $1 × 10^6$ cells per ml. At day 14, the cultures contained >90% bone marrow derived eosinophils (BMDEs), indicated by high side scatter profile and expression of Siglec-F.

### In vitro engulfment assay of senescent cells

Cells were plated and senescence was induced as described above. Human senescent cells were labeled with Cytolight Rapid dye (3 μM for 20 min at 37°C; 4705; Sartorius) according to the manufacturer's guidelines. Differentiated macrophages were added in specified ratios (between macrophages and senescent cells as indicated in the figure legends). Plates were immediately subjected to the IncuCyte device for live cell imaging for 48 h. Confluency of cells, fluorescence intensities/area/object count were developed with the IncuCyte software.

### Apoptotic corpse generation and labeling

To generate apoptotic corpses (AC, prey cells, Jurkat, Raji, or TK-1 cells) were transferred into 10 cm petri dishes (8–10 ml of cell suspension with $1 \times 10^6$ cells per ml) and exposed to UV light for 5–10 min. Apoptosis induction was confirmed by Annexin V (Sartorius; 4661), Caspase 3/7 (Sartorius; 4704), and Cytotox (Sartorius; 4633) staining over time. For live time imaging (IncuCyte) of uptake of apoptotic corpses by macrophages, apoptotic Raji, Jurkat, and TK-1 cells were labeled with pHrodo dye (250 ng/ml) according to manufacturer's protocol (Sartorius; 4649). After labeling, corpses were added to macrophages in specified ratios (between macrophages and ACs as indicated in the figure legends).

### Efferocytosis assays by IncuCyte S3

Cells were plated in 96-well plates (Nunc; 187006) and senescence was induced as described above. Matching non-senescent (proliferating; $4 \times 10^3$ cells per well) control cells were seeded as well. After 4–6 h differentiated macrophages were collected by gently scraping and were subsequently added to the 96 well plates with senescent or proliferating cells (MDM: 15,000; BMDM: 20,000). These macrophage co-cultures were rested overnight (16 h) before adding pHrodo-labeled Jurkat or Raji cells to macrophages in specified ratios as indicated in the figure legends. Corpse engulfment was monitored using the IncuCyte S3 system taking images every hour for 24 h in an incubator at 37°C. The first image time point was generally acquired within 30 min of co-culture using a 10× objective at 300 ms exposures per field. Image analysis was carried out using IncuCyte 2021B software calculating the total integrated intensity (RCUxμm²/Image) corresponding to the efferocytotic activity of macrophages.

### Efferocytosis assays by immunofluorescence

Cells were plated in either 96-well plates (for Endpoint IncuCyte measurements; 6055302; Perkin Elmer) or 8-chamber slides (for Confocal microscopy; $8 \times 10^3$ cells per well; 80826; ibidi), and senescence was induced as described above. Matching non-senescent (proliferating; $4 \times 10^3$ cells per well) control cells were seeded as well. After 4–6 h differentiated macrophages were collected by gently scraping and were subsequently added to the wells with senescent or proliferating cells (96-well: MDM: 15,000; slide: 30,000). These macrophage co-cultures were rested overnight (16 h) before adding apoptotic Raji cells to macrophages in specified ratios as indicated in the figure legends. Rajis were co-labeled in HBSS with Cell-tracker green (1:

1,000) for 45 min and CypHer5E (1 μg/ml) for 30 min at 37°C. 1, 2 or 4 h post exposure to apoptotic cells, co-cultures are washed twice with PBS (14190-094; Gibco) to remove un-engulfed Raji cells. Afterwards, cells were fixed for 15 min with 4% Paraformaldehyde (PFA). Afterwards, cells were washed again three times with PBS, permeabilized for 20 min with 0.1% Triton X-100 in PBS and blocked (3% BSA in PBS) for 30 min. Cells were incubated over night at 4°C with primary antibodies to detect CD45 (1:500; MA5-17687; Invitrogen) which were diluted in 1% BSA in PBS. Subsequently, cells were washed three times with PBS and secondary antibodies (diluted in PBS; 1:500; anti-rat Alexa 568; A11077; Invitrogen) containing Hoechst staining (H3570; 1:500) was added for 60 min at room temperature in the dark. Cells were washed three times with PBS and kept in PBS at 4°C until imaging. Slides were imaged using a LSM710 Confocal Laser Scanning Microscope. 96-well plates were imaged using the IncuCyte S3 system using a 20× objective at 300 and 400 ms exposures per field. Image analysis was carried out using IncuCyte 2021B software calculating the total object count of green and overlap events per image corresponding to the efferocytotic activity of macrophages.

### Efferocytosis assays by flow cytometry

Senescent cells were plated in 12-well plates as described before. 24 h prior to the assay, the culture medium was changed. For co-culture conditions BMDMs were added to the senescent cells. Senescent cells and macrophages were co-cultured for 6 and 24 h. PKH26-labeled apoptotic corpses (ACs) were incubated with the macrophages for 60 min (if not stated otherwise) at a 1:5 ratio (macrophages: AC) followed by washing three times with PBS to remove unbound ACs. The co-culture was analyzed by flow cytometry. Cells were suspended in FACS buffer (PBS containing 1% FBS) and incubated with $F_c$ block and cell surface markers for 30 min on ice in the dark. Afterwards cells were washed in FACS buffer and suspended for analysis on a flow cytometer. Data analysis was carried out using FlowJo software.

### Immunostaining

Cells were seeded in 8-chamber slides ($8 \times 10^3$ cells per well; 80826; ibidi), and senescence was induced as described above. Proliferating control cells were seeded the day before fixation ($8 \times 10^3$ cells per well). Cells were washed two times with PBS (14190-094; Gibco) and fixed for 15 min with 4% PFA. Afterwards, cells were washed again three times with PBS, permeabilized for 20 min with 0.1% Triton X-100 in PBS and blocked (3% BSA in PBS) for 30 min. Cells were incubated for 1.5 h at 37°C with primary antibodies to detect CD47 (1:40; AF4670; R&D), CD24 (1:30; MA5-11828; Invitrogen), CD45 (1:500; MA5-17687; Invitrogen) and SHP1 (MA5-11669; human:1:100; Invitrogen; PA5-27803; mouse: 1:200; Thermo Fisher Scientific) which were diluted in 1% BSA in PBS. Subsequently, cells were washed three times with PBS and secondary antibodies (diluted PBS; anti-sheep Alexa Fluor 488, A11015; Invitrogen; anti-mouse Alexa 488, A32723; Invitrogen; anti-rat Alexa 647, A21247; Invitrogen) containing Hoechst staining (H3570; 1:500) was added for 30 min at 37°C in the dark. Phalloidin staining was performed according to manufacturer's instructions (8878; Cell

Signaling). Cells were washed three times with PBS and kept in PBS at 4°C until imaging.

## Fluorescence microscopy

Fluorescence microscopy was performed using a LSM710 or LSM780 Confocal Laser Scanning Microscope (Zeiss) at room temperature equipped with a 20× water-immersion (Plan-Apochromat" 20×/0,8 M27, Zeiss), a 40× water-immersion (Objective LD C-Apochromat 40×/1.1 W Corr M27, Zeiss) or a 40× oil-immersion (Plan-Apochromat 40×/1,4 Oil DIC, Zeiss, at LSM 780) objective lens. Images were acquired using ZEN Black software (Zeiss), final adjustments for brightness were done using either ZEN blue software (Zeiss) or ImageJ (mouse SHP1 staining). All images can be made available upon request.

## Crispr/Cas9-mediated genome engineering

Primers for guide RNAs (gRNA) for murine CD47 were designed using web-based tools from the Broad Institute (Cambridge, Mass). Phosphorylated primer oligomers were annealed, and the product was ligated into pSpCas9(BB)-2A-GFP (PX458) that was previously digested with BbsI and the correct plasmids were transfected into WT NIH-3T3 cells through Lipofectamine. The GFP positive subpopulation was separated from the bulk population by flow cytometry. The sorted bulk population was diluted to grow single cell clones. These clones were checked for gene loss by flow cytometry using a CD47 antibody. To generate Panc1 CD47 knockout cells, WT Panc1 cells were transfected by electroporation with a gRNA-Cas9 complex containing TrueGuide Synthetic sgRNA targeting human CD47, TrueGuide tracrRNA and TrueCut Cas9 Protein v2 (Invitrogen). After 2 wk, single cells were seeded and clonally expanded. CD47 knockout clones were selected based on lack of staining with anti-CD47 (B6H12) using flow cytometry. Gene disruption was validated by sequence analysis of the relevant gene locus by TIDE analysis.

## RNA isolation and analysis

Cells were washed with PBS and subsequently lysed in RLT Plus Buffer (1053393; Qiagen) followed by RNA isolation according to the manufacturer's instructions (MagMAX-96 Total RNA Isolation Kit, AM1830, Thermo Fisher Scientific). RNA quality and concentration were determined by absorbance at 260 and 280 nm using a NanoDrop spectrophotometer (Thermo Fisher Scientific). Complementary DNA was synthesized using High-Capacity cDNA Reverse Transcription Kit (4368813; Thermo Fisher Scientific) followed by quantitative reverse transcription polymerase chain reaction. p21, IL6, MKI67, CXCL1, CD47, QPCT and QPCTL TaqMan gene expression assays were purchased from Thermo Fisher Scientific. Ct values were normalized to Ct values of HPRT1 housekeeping control gene to generate delta Ct values. To obtain delta/delta Ct values, delta CT values were normalized to the respective proliferating control. Subsequently, fold change was calculated using the following formula: (2^(-delta/deltaCt)).

## Immunoblotting

Cell lysates were prepared on ice with RIPA buffer (R0278; Sigma-Aldrich) substituted with protease and phosphatase inhibitors (1861284; Thermo Fisher Scientific). Proteins were separated by 4–12% gradient SDS-PAGE in Tris-HCl buffer and transferred to nitrocellulose membranes. Equal transfer was verified with Ponceau Red stain (P7170-1L; Sigma-Aldrich). Membranes were blocked in 5% nonfat milk (42590.10; Serva) and probed with primary antibodies overnight at 4°C. Anti-P21 (2947S) was obtained from Cell Signaling Technology. Anti-CD47 (AF4670) antibody was purchased from R&D Systems. Anti-CD22 (ab218340) and anti-CD24 (ab179821) were obtained from Abcam. As a loading control, anti-GAPDH (2118) antibody was used from Cell Signaling Technology. Membranes were washed and goat anti-rabbit IgG-horseradish peroxidase (1:2,500; RPN4301; Amersham), goat anti-mouse IgG-horseradish peroxidase (1:1,500; A8924; Sigma-Aldrich) or donkey anti-sheep IgG-horseradish peroxidase (1:4,000; A3415; Sigma-Aldrich) were used as secondary antibody. Membranes were developed using chemiluminescence reagents (NEL104001EA; Perkin Elmer) and the X-Stella system (Reytest Isotope Messgeraete GmbH). Image analysis was performed using the AIDA Imaging analyzer (Reytest Isotope Messgeraete GmbH).

## Tabula muris senis analysis

We extracted cell type annotation and read counts matrix as anndata object from the tabula muris senis droplet processed data set using Scanpy (version 1.7.2; Wolf et al., 2018). The data was normalized using total normalization and scaled using scaling method implemented in Scanpy after removing genes with minimum counts of 10. Scaled read counts were used to generate gene expression plot across different ages.

## Re-analysis of RNA-seq data sets

RNASeq data (FASTQ files) for rat liver fibrosis models has been downloaded from the ArrayExpress database at EMBL-EBI (www.ebi.ac.uk/arrayexpress) under accession numbers "E-MTAB-10546 - A time-course study of bile duct ligation (BDL)-induced liver fibrosis in rats" (https://www.ebi.ac.uk/arrayexpress/experiments/E-MTAB-10546/) and "E-MTAB-10547 - A time-course study of thioacetamide (TAA)-induced liver fibrosis in rats" (https://www.ebi.ac.uk/arrayexpress/experiments/E-MTAB-10547/). Sequencing reads (FASTQ files) from the RNA-seq data were processed with a pipeline building upon the implementation of the ENCODE "Long RNA-seq" pipeline and filtered reads were mapped against the *Rattus norvegicus* genome (Rnor_6.0) using the STAR (v2.5.2b) aligner (https://doi.org/10.1093/bioinformatics/bts635) allowing for soft clipping of adapter sequences. For quantification, we used transcript annotation files from Ensembl version 86. Gene expression levels were quantified with the above annotations, using RSEM (v1.3.0; Li and Dewey, 2011) and featureCounts (v1.5.1; Liao et al., 2014). Quality controls were implemented using FastQC (v0.11.5; http://www.bioinformatics.babraham.ac.uk/projects/fastqc/, picardmetrics [v0.2.4], and https://github.com/slowkow/picardmetrics) and dupRadar (v1.0.0; Sayols et al., 2016) at the respective steps. Differential expression analysis was performed on the mapped counts derived from featureCount using limma/voom (Ritchie et al., 2015). If not otherwise stated, we used an absolute $\log_2$ fold change cut-off of 0.5 and a false discovery rate (FDR) of <0.05.

## Proteomic analysis

Cells were lysed in SDS lysis buffer (2% SDS in 100 mM TEAB), boiled (15 min, 95°C, 500 rpm) and sonicated in a sonication bath (10 min). DNA was sheared with 0.5 µl Benzonase Nuclease (71205-25KUN; Merck Millipore) and 2 mM MgCl2 at 37°C (30 min, 500 rpm). Protein amount was quantified using the Pierce BCA Protein assay (Thermo Fisher Scientific). Reversibly oxidized cysteines were reduced with 10 mM tris-(2-carboxyethyl)-phosphine followed by alkylation with 20 mM iodoacetamide (each step 30 min, 22°C, 600 rpm, in the dark). Proteins were precipitated with five volumes of ice-cold acetone (1 h, −20°C). Following centrifugation (20 min, 4°C, 13,000 rpm) protein pellets were washed with 80% acetone in a sonication bath (10 min). After another centrifugation step the protein pellet was resuspended in 100 µl 100 mM TEAB. Proteins were digested overnight with LysC/trypsin (1:50 protease:protein ratio, 37°C, 600 rpm) followed by acidification to a final concentration of 1% trifluoroacetic acid (TFA). After centrifugation (5 min, 13,000 rpm) peptides were desalted by solid-phase extraction using Oasis HLB cartridges (10 mg; Waters) as described previously (Griesser et al., 2020), resuspended in 3% ACN/0.1% TFA and quantified using the CBQCA assay (Thermo Fisher Scientific). Peptides were separated on an UltiMate 3000 RSLCnano system and analyzed on an Orbitrap Eclipse Tribrid mass spectrometer in combination with the FAIMS Pro Interface (Thermo Fisher Scientific). Peptides were loaded onto a 100 µm × 2 cm Acclaim PepMap C18 trap column (5 µm, 100 Å) for 6 min with 3% ACN/0.1% TFA and a constant flow of 10 µl/min and separated on a 75 µm × 75 cm EASY-Spray C18 column (2 µm, 100 Å; Thermo Fisher Scientific) at 50°C with a flow rate of 300 nl/min. Solvents used were 0.1% formic acid (A) and 80% ACN/0.1% formic acid (B). Samples were separated over a multi-step gradient from 8% to 16% B in 40 min, to 25% in 25 min and 35% in 18 min with a total measurement time of 120 min. The spray was initiated by applying 2.1 kV to the EASY-Spray emitter. The ion transfer capillary temperature was set to 275°C and the radio frequency of the ion funnel to 30%. Data were acquired in data-dependent mode using advanced peak detection. The FAIMS Pro interface was used in standard resolution mode without gas flow and two compensation voltages (−40 V for 1.2 s and −60 V for 0.8 s). The full scan was acquired in the orbitrap (m/z 375-1,500, mass resolution 120,000, normalized automatic gain control [AGC] target 100%, maximum injection time [IT] 50 ms). Only precursor ions with charges between 2 and 6, and a minimum intensity of 5.0E+03 were selected for fragmentation using higher-energy collision dissociation in the ion trap with 28% collision energy. MS2 spectra were acquired with a "rapid" ion trap scan rate (normalized AGC target 100%, dynamic maximum IT) using a dynamic exclusion of 45 s. Acquired tandem mass spectra were searched against the UniProtKB/Swiss-Prot human database (downloaded in September 2021, containing 20,386 protein sequences) using the Sequest search engine in Proteome Discoverer 2.4 (Thermo Fisher Scientific). Trypsin was specified as the cleavage enzyme, allowing up to two missed cleavages, and mass tolerances of 10 ppm and 0.6 Da were selected for precursor and fragment ions, respectively. Carbamidomethylation (Cys) was used as fixed modification, and oxidation (Met), deamidation (Asn, Gln), N-terminal Gln to pyro-Glu conversion and acetylation (protein N-terminus) were set as variable modifications. Using the percolator node, a decoy-based false discovery rate (FDR) of 1% was applied to peptide-spectrum matches. Additionally, an FDR of 1% was applied to peptide and protein identifications (high confidence). Only unique peptides were used for quantification of precursor ions and a normalization based on the total peptide amount was applied to address differences in sample loadings. Contaminants were excluded and only master proteins with high confidence identified with minimum two unique peptides were considered for further analysis. For the NHLF dataset another normalization step was applied to address donor variability. Therefore, the median of protein intensities was calculated for each donor and protein and used to calculate normalization factors. Mass spectrometry raw files from proteomic analysis have been deposited to the ProteomeXchange Consortium (Vizcaíno et al., 2014) through the MassIVE partner repository with the dataset identifier MSV000090607.

## β-Galactosidase staining and BrdU incorporation assays

SA-β-Gal staining was performed using a SA-β-Gal staining Kit (9860; Cell signeling Technology) according to the manufacturer's instructions. Senescent cells were identified as blue-stained cells under light microscopy (AxioVert 135T; Zeiss). BrdU incorporation assays (11647229001; Roche) was performed according to the manufacturer's guidelines.

## Cytokine analysis

A Meso Scale Discovery multiplex cytokine immunoassay panel was used to quantitate secreted components of the senescence associated secretory phenotype (SASP), including IL6 and CXCL1 according to manufacturer's instructions (Mesoscale Diagnostics). Human CXCL1 quantification was performed by ELISA measurements according to the manufacturer's guidelines (DY275; R&D Systems).

## SIRPα binding reporter assay

Cells were plated in opaque 96-well plates (4 × 10³ cells per well, 136101; Nunc) and senescence was induced as described above. Matching non-senescent (proliferating; 6 × 10³ cells per well) control cells were seeded as well. The next day SIRPα binding was quantified by employing an SIRPα PathHunter Bioassay Detection Kit (93-0933; Eurofins), which was co-developed with Eurofins. The assay was performed according to the manufacturers protocol. Medium was aspirated and reporter cells (5 × 10⁴ cells per well) were added to the cells.

## QPCT/L enzyme activity assay

Cells were grown in 6-well plates and lysed with MSD Lysis buffer (Mesoscale Discovery, R60TX-2) substituted with protease and phosphatase inhibitors (1861284; Thermo Fisher Scientific). Proteins were quantified using Pierce BCA Protein Assay Kit (23225; Thermo Fisher Scientific). QPCT/L Enzyme activity was investigated by performing SensoLyte Green Glutaminyl Cyclase Activity Assay Kit (AS72230; AnaSpec) according to the manufacturer's guidelines using 25 µg of protein input. No statistical analysis was performed.

### Treatment with QPCT/L inhibitor (SEN177)

Fibroblasts were pre-treated for 48 h with the QPCT/L inhibitor SEN177 (1, 5, 10 μM; SML1615; Sigma-Aldrich), then cells were plated in 96-well plates ($4 \times 10^3$ cells per well, 187006; Nunc) and senescence was induced as described above in the presence of SEN177. SEN177 was added every other day to the cells. Matching non-senescent (proliferating; $4 \times 10^3$ cells per well, pre-treated for 48 h with SEN177) control cells were seeded as well.

### Statistical analysis

The significance of differences in the experimental data were determined using GraphPad Prism software. All data involving statistics are presented as mean ± SEM. The number of replicates and the statistical test used are described in the figure legends. When employing Student's $t$ test, data distribution was assumed to be normal, but this was not formally tested. Statistical analysis of the Tabula Muris Senis data set was performed using Wilcoxon test. Statistical significance in the reanalyzed RNAseq rat liver fibrosis datasets was determined by limma/voom (Law et al., 2014; Ritchie et al., 2015) based on counts derived from featureCounts (Liao et al., 2014).

### Online supplemental material

Fig. S1 shows that activated macrophages (mouse and human) do not remove co-cultured senescent cells (labeled with Green Cytolight rapid Dye) over time, while the capability to engulf apoptotic corpses is impaired. Fig. S2 shows that senescent cells inhibit macrophage's ability to phagocytose. Fig. S3 shows increased CD24 expression of senescent epithelial cells. Fig. S4 shows increased expression of QPCT and QPCTL in senescent cells. Video 1 shows tdTomato+ BMDMs in direct co-culture with senescent actin-GFP mouse embryonic fibroblasts (MEFs). Table S1 lists all materials (e.g. chemicals, antibodies, primers) used in this study.

## Acknowledgments

We thank members of the P.J. Murray and K.C. El Kasmi groups for discussions and comments on the manuscript. We acknowledge the contributions of the Max Planck Institute of Biochemistry Imaging Core Facility and Eva-Maria Eckl for contributions and insights in the early phase of this project. We thank Karin Fiesel (Boehringer Ingelheim, Biberach) for the curation of the published RNA-seq datasets. We also thank Bettina Gercken (Institute for Clinical Chemistry and Laboratory Medicine, Technische Universität Dresden) for technical assistance.

Work in the PJM laboratory is supported by a research contract from Boehringer Ingelheim encompassing macrophage biology in disease, FOR-2599 (type 2 tissue immunity), SFB-TRR 127 (Xenotransplantation) from the Deutsche Forschungsgemeinschaft (DFG) and the Max-Planck-Gesellschaft. The work in the TC laboratory is supported by the SFB-TRR 127 (Xenotransplantation) from the DFG. Open Access funding provided by the Max Planck Society.

Author contributions: D. Schloesser investigation, Visualization, Writing – original draft, Writing – review & editing, Conceptualization, Formal analysis, Methodology, Validation, Data curation. L. Lindenthal Investigation, Visualization, Formal analysis, Methodology, Data curation. J. Sauer Investigation. K-J. Chung Resources, Investigation. T. Chavakis Resources Investigation, Writing – review & editing. E. Griesser Investigation, Formal analysis. P. Baskaran Formal analysis, Data curation. U. Maier-Habelsberger Investigation. K. Fundel-Clemens Formal analysis, Data curation. I. Schlotthauer Methodology. C.K. Watson Writing – review. L.K. Swee Writing – review & editing. F. Igney Writing – review & editing, M.S. Huber-Lang Writing – review & editing. J.E. Park Writing – review & editing. M-J. Thomas Writing – review & editing. K.C. El Kasmi Conceptualization, Supervision, Writing – original draft, Writing – review & editing, Funding acquisition, Project administration. P.J. Murray Conceptualization, Supervision, Writing – original draft, Writing – review & editing, Funding acquisition, Project administration.

Disclosures: D. Schloesser, J. Sauer, E. Griesser, U. Maier-Habelsberger, K. Fundel-Clemens, I. Schlotthauer, C. Watson, F. Igney, M-J. Thomas, and K.C. El Kasmi are employees of Boehringer Ingelheim Pharma at which part of the current work was performed and conceptualized. T. Chavakis reported grants from Deutsche Forschungsgemeinschaft and grants from European Research Council during the conduct of the study. P. Murray is a member of the scientific advisory boards of Palleon Pharmaceuticals and ImCheck Pharma. These relationships have no bearing or relevance to the current work. As noted in the manuscript, the present work is the result of a close collaboration between his group and the researchers at Boehringer Ingelheim, which is funded in part by a cooperation contract (as stated). No other disclosures were reported.

Submitted: 20 July 2022

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

# Supplemental material

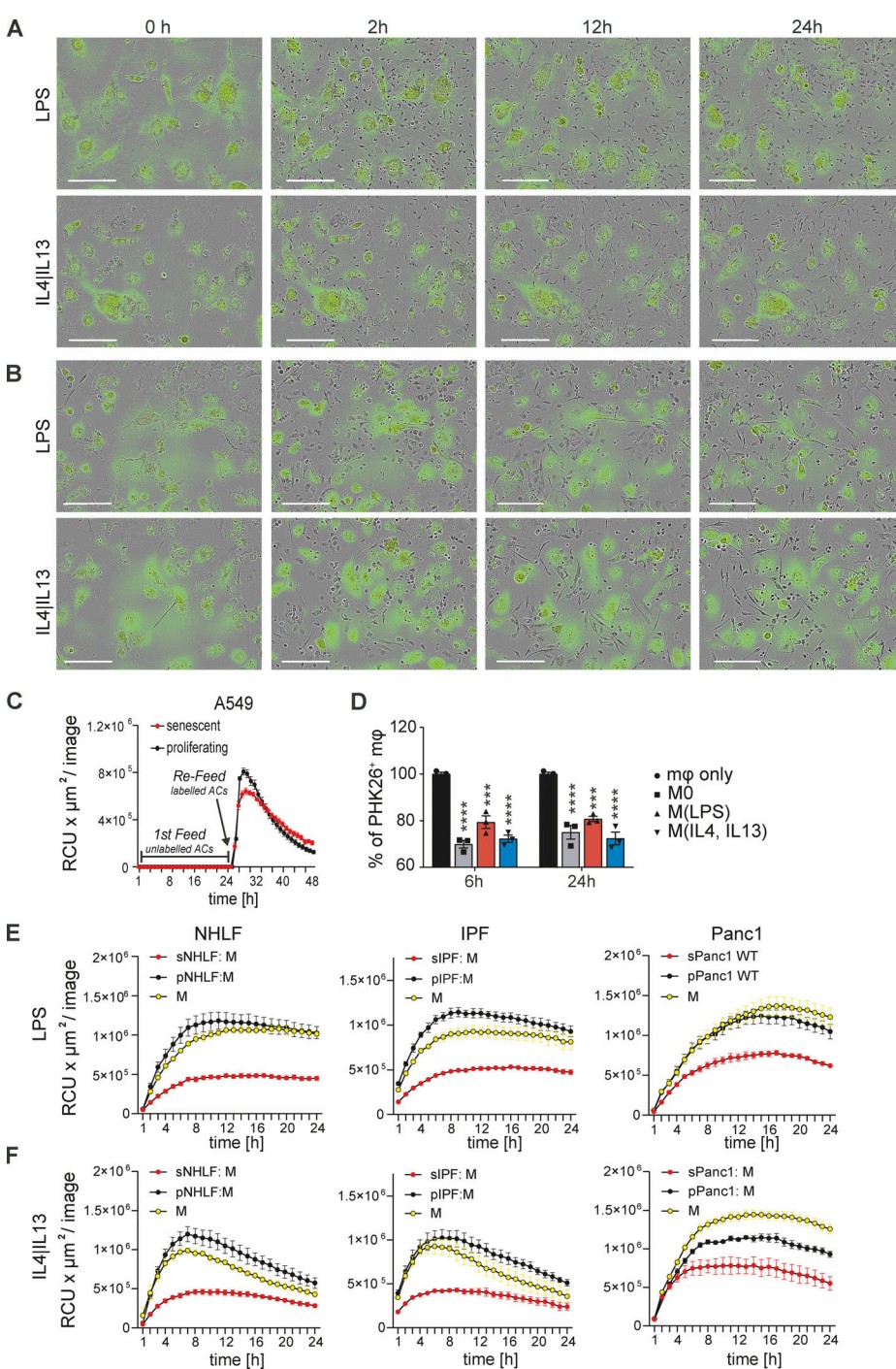

Figure S1. **Impairment of efferocytosis by senescent cells is independent of macrophage polarization. (A)** Senescent 3T3 cells (labeled with Green Cytolight Rapid Dye) were co-cultured with naïve BMDMs. Co-cultures were either stimulated with LPS (upper panel) or IL4 and IL13 (lower panel). Images were taken over 24 h. In each image group, the same field of view was shown across time. Data are representative of three independent experiments; scale bars indicate 200 μm. **(B)** Senescent Panc1 cells (labeled with Green Cytolight Rapid Dye) were co-cultured with naïve MDMs. Co-cultures were either stimulated with LPS (upper panel) or IL4 and IL13 (lower panel). Images were taken over 24 h. In each image group, the same field of view was shown across time. Data are representative of three independent experiments; scale bars indicate 200 μm. **(C)** Quantification of efferocytosis of apoptotic corpses by MDMs in the presence of proliferating (black line) or senescent (red line) A549 cells over time. First, co-cultured macrophages were exposed to unlabeled apoptotic corpses. After 24 h, pHrodo labeled apoptotic corpses were added to the co-culture and efferocytotic activity of the macrophages was monitored over time by the IncuCyte S3 system. **(D)** Efferocytosis of apoptotic corpses by differently stimulated macrophages in the presence of senescent cells (induced by palbociclib). Unstimulated macrophages were polarized to a M1 phenotype by the addition of LPS or to a M2 phenotype by the addition of IL4 and IL13. Data are representative of three independent experiments. All values are means ± SEM; ***P < 0.001; ****P < 0.0001. Statistically significant differences were determined by one-way ANOVA with Bonferroni correction; $n = 3$ biological replicates. **(E)** Quantification of efferocytosis of apoptotic corpses by MDMs that were co-cultured with NHLF, IPF or Panc1 cells and then further stimulated with either LPS (upper row) or IL4 and IL13 (lower row) Efferocytotic capability of macrophages in single (yellow line) or co-culture with proliferating (black line) or senescent (red line) was monitored over time.

Figure S2. **Senescent cells inhibit macrophages' ability to phagocytose. (A)** Schematic overview of the experimental design to monitor phagocytosis by macrophages in co-culture with proliferating or senescent cells. 16 h after the assembly of the co-cultures, either *Escherichia coli*, secondary necrotic corpses or inert silicon beads were added. Uptake by macrophages was monitored over time using the IncuCyte S3 system. **(B)** Removal of the labeled phagocytotic prey by macrophages was quantified over time. Macrophages were in co-culture with either proliferating (black line) or senescent cells (red line) of the indicated origin.

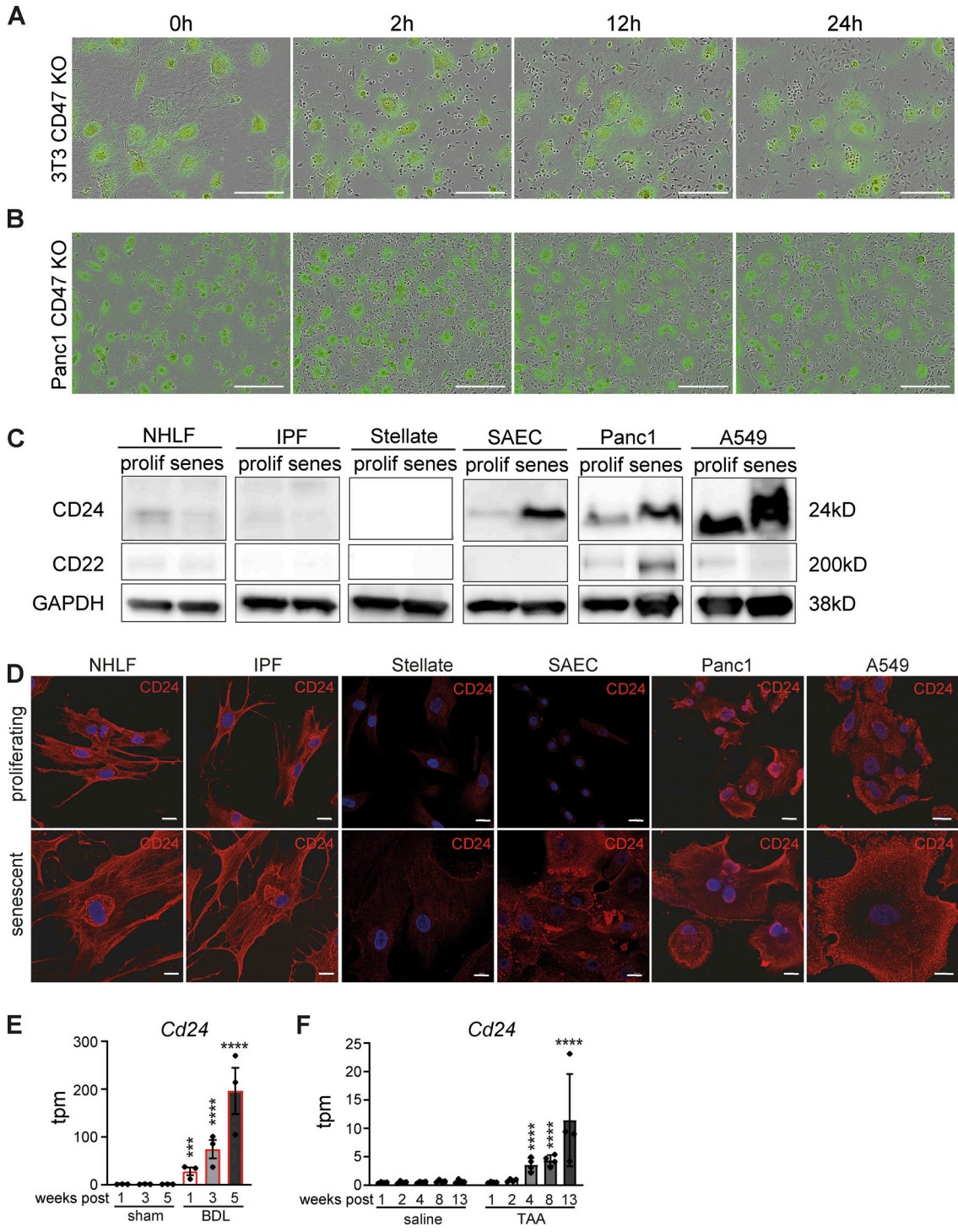

Figure S3. **Increased CD24 expression in senescent epithelial cells. (A)** Senescent CD47 KO 3T3 cells (labeled with Green Cytolight Rapid Dye) were co-cultured with naïve BMDMs (unstained). Images were taken over 24 h. In each image group, the same field of view was shown across time. Data are representative of three independent experiments; scale bars indicate 200 µm. **(B)** Senescent CD47 KO Panc1 cells (labeled with Green Cytolight Rapid Dye) were co-cultured with naïve MDMs (unstained). Images were taken over 24 h. In each image group, the same field of view was shown across time. Data are representative of three independent experiments; scale bars indicate 200 µm. **(C)** Whole-cell lysates from indicated cell types were isolated and analyzed by Western blotting for the indicated proteins. Senescence was induced by chemical treatment. Data are representative of at least two independent experiments. GAPDH images are derived from the same blot as Fig. 7, C and E (SAEC) and Fig. 8 G. CD24 image is derived from the same blot as in Fig. 8 G. **(D)** Representative immunofluorescence images from indicated cell types (proliferating or senescent) stained for CD24 (red) and Hoechst (blue). Scale bars indicate 20 µm. n = 2. **(E and F)** Expression levels obtained from RNA-sequencing data for liver fibrosis rat models (Wang et al. 2021). Transcripts per million (tpm) expression values for *Cd24* in rat livers derived from bile duct ligation (BDL; E) or thioacetamide (TAA)-induced liver fibrosis (F). Data are represented as mean ± SEM. Significant de-regulation (***P < 0.0005, ****P < 0.0001) was determined by limma/voom based on counts derived from featureCounts. Source data are available for this figure: SourceData FS3.

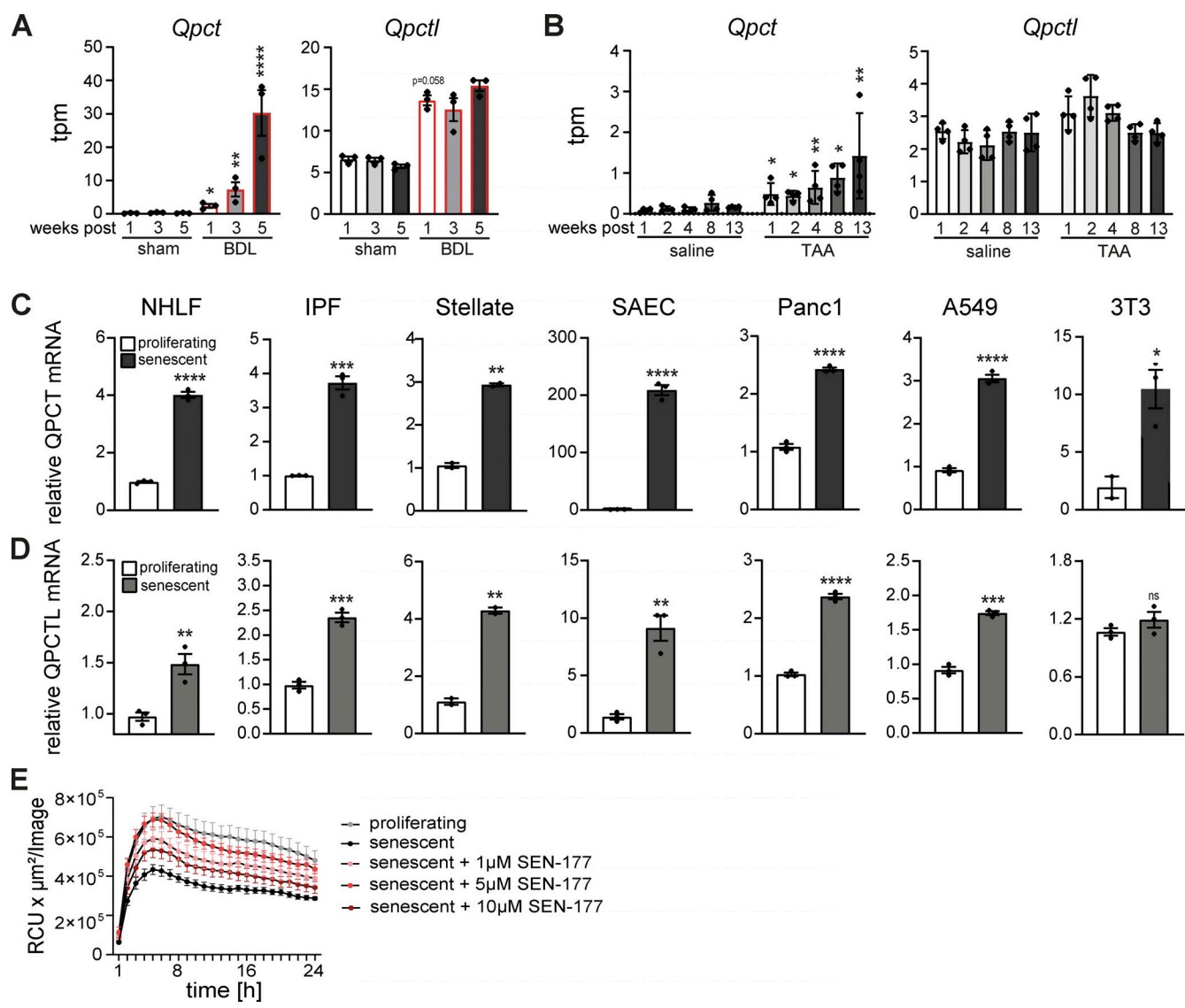

Figure S4.   **Increased QPCT/L expression in senescent cells. (A and B)** Expression levels obtained from RNA-sequencing data for liver fibrosis rat models (Wang et al. 2021). Transcripts per million (tpm) expression values for *Qpct* and *Qpctl* in rat livers derived from BDL (A) or TAA-induced liver fibrosis (B). Data are represented as mean ± SEM. Significant de-regulation (*P < 0.05, **P < 0.005, ****P < 0.0001) was determined by limma/voom based on counts derived from featureCounts. **(C and D)** Quantification the relative *QPCT* (C) and *QPCTL* (D) mRNA levels by qPCR. Expression levels of senescent cells were normalized to the respective proliferating control (white bar). Data are representative of three independent experiments. All values are means ± SEM. *P < 0.05, **P < 0.005, ***P < 0.0005, ****P < 0.0001. Statistically significant differences were determined by unpaired Student's *t* test. **(E)** Quantification of efferocytosis of apoptotic corpses in co-cultures of human MDMs and proliferating (gray bar) or senescent IPF-derived lung fibroblasts. Senescent IPF-derived lung fibroblasts were treated with the QPCTL inhibitor SEN-177 (1, 5, 10 µM) prior the assembly of the co-culture with MDMs. Efferocytotic capability of macrophages was monitored over time using the IncuCyte S3 system. Data are representative of three independent experiments. All values are means ± SEM.

Video 1.   **Macrophages remain motile in direct co-culture with senescent cells.** Senescence was induced in actin-GFP mouse embryonic fibroblasts (MEFs) by passaging stress until the Hayflick limit was reached. Senescent MEFs were co-cultured with BMDMs, isolated from tdTomato+ mice, in a ratio of 10 BMDMs to 1 senescent cell and imaged over 48 h every 20 min using the IncuCyte S3 system. Data are representative of three independent experiments; scale bars indicate 300 µm.

**Provided online is Table S1. Table S1 shows lists all materials (e.g. chemicals, antibodies, primers) used in this study.**

