## [Peer Review File · The Journal of Cell Biology]

Senescent cells suppress macrophage-mediated corpse removal via upregulation of the CD47-QPCT/L axis

Daniela Schloesser, Laura Lindenthal, Julia Sauer, Kyoung-Jin Chung, Triantafyllos Chavakis, Eva Griesser, Praveen Baskaran, Ulrike Maier-Habelsberger, Katrin Fundel-Clemens, Ines Schlotthauer, Carolin Watson, Lee Swee, Frederik Igney, John Park, Markus Huber-Lang, Matthew-James Thomas, Karim El Kasmi, and Peter Murray

Corresponding Author(s): Peter Murray, Max Planck Institute of Biochemistry and Karim El Kasmi, Boehringer Ingelheim

Review Timeline:	Submission Date:	2022-07-20
	Editorial Decision:	2022-08-26
	Revision Received:	2022-10-19
	Editorial Decision:	2022-10-26
	Revision Received:	2022-11-01

Monitoring Editor: Jennifer Stow

Scientific Editor: Lucia Morgado-Palacin

Transaction Report:

DOI: <https://doi.org/10.1083/jcb.202207097>

August 26, 2022

Re: JCB manuscript #202207097

Prof. Peter Murray
Max Planck Institute of Biochemistry
Am Klopferspitz 18
Martinsried 82152
Germany

Dear Prof. Murray,

Thank you for submitting your manuscript entitled "Senescent cells suppress macrophage-mediated corpse removal via upregulation of the CD47-QPCT/L axis". The manuscript was assessed by expert reviewers, whose comments are appended to this letter. We invite you to submit a revision if you can address the reviewers' key concerns, as outlined here.

As you will see, the reviewers are enthusiastic about this work. The minor comments of all three reviewers should be addressed since most pertain to improving/correcting data representation. Reviewer #3 suggests two new experiments and the valid points raised here should be addressed with discussion of these points and with the inclusion of new data where possible, especially to show macrophage activity at earlier time points. We hope that you will be able to address each of reviewer #1's other points, though.

GENERAL GUIDELINES:

Text limits: Character count for an Article is < 40,000, not including spaces. Count includes title page, abstract, introduction, results, discussion, and acknowledgments. Count does not include materials and methods, figure legends, references, tables, or supplemental legends.

Figures: Articles may have up to 10 main text figures. Figures must be prepared according to the policies outlined in our Instructions to Authors, under Data Presentation, <https://jcb.rupress.org/site/misc/ifora.xhtml>. All figures in accepted manuscripts will be screened prior to publication.

Supplemental information: There are strict limits on the allowable amount of supplemental data. Articles may have up to 5 supplemental figures. Up to 10 supplemental videos or flash animations are allowed. A summary of all supplemental material should appear at the end of the Materials and methods section.

Please note that JCB now requires authors to submit Source Data used to generate figures containing gels and Western blots with all revised manuscripts. This Source Data consists of fully uncropped and unprocessed images for each gel/blot displayed in the main and supplemental figures. Since your paper includes cropped gel and/or blot images, please be sure to provide one Source Data file for each figure that contains gels and/or blots along with your revised manuscript files. File names for Source Data figures should be alphanumeric without any spaces or special characters (i.e., SourceDataF#, where F# refers to the associated main figure number or SourceDataFS# for those associated with Supplementary figures). The lanes of the gels/blots should be labeled as they are in the associated figure, the place where cropping was applied should be marked (with a box), and molecular weight/size standards should be labeled wherever possible. Source Data files will be made available to reviewers during evaluation of revised manuscripts and, if your paper is eventually published in JCB, the files will be directly linked to specific figures in the published article.

The typical timeframe for revisions is three to four months. While most universities and institutes have reopened labs and allowed researchers to begin working at nearly pre-pandemic levels, we at JCB realize that the lingering effects of the COVID-19 pandemic may still be impacting some aspects of your work, including the acquisition of equipment and reagents. Therefore,

if you anticipate any difficulties in meeting this aforementioned revision time limit, please contact us and we can work with you to find an appropriate time frame for resubmission. Please note that papers are generally considered through only one revision cycle, so any revised manuscript will likely be either accepted or rejected.

Thank you for this interesting contribution to Journal of Cell Biology. You can contact us at the journal office with any questions, cellbio@rockefeller.edu.

Sincerely,

Jennifer Stow
Monitoring Editor
Journal of Cell Biology

Lucia Morgado-Palacin, PhD
Scientific Editor
Journal of Cell Biology

Reviewer #1 (Comments to the Authors (Required)):

Schloesser, Lindenthal and colleagues report on a novel mechanism by which senescent cells suppress the ability of macrophages to engulf bystander apoptotic cells. Specifically, the authors described a "cell-mediated efferocytosis suppression" mechanisms wherein senescent cells interact, in a cell-cell contact manner, with macrophages via the CD47 (and CD24)-SIRPa axis to limit the ability of the phagocyte to engulf apoptotic cargo. Interestingly, they also extended these findings to the uptake of bacterial, necrotic cargo and beads.

Significant strengths of the manuscript are the biological process being studied and the rigorous and methodic approach. The manuscript is clearly written, and the data is compelling. These thorough in vitro studies should inspire future efforts to test this newly described mechanism in vivo.

This reviewer finds that some statements should be reconsidered and revised appropriately.

The authors state that: "Blocking CD47 (Figure 4D) on human senescent cells (primary lung fibroblasts derived from healthy (Figure 4E) or IPF-lungs (Figure 4F)) using anti-CD47 FAB fragments augmented corpse removal by co-cultured MDMs relative to co-cultures without CD47 blockade." This is not clear from the data presented, as there seems to be no significant differences in the AUC between untreated and anti-CD47 treated conditions.

NSC87877 is both a potent SHP-1 and SHP-2 inhibitor. It would be important for the readers that this is clarified.

The manuscript is already very comprehensive. It would have been ideal to complement the SHP-1 immunolocalization studies with biochemical approaches on its association with SIRPa. A discussion on this would be useful for the readers as well.

Reviewer #2 (Comments to the Authors (Required)):

In the present manuscript Schloesser et al. investigate the effect of senescent cells on homeostatic macrophage function. The study shows that senescent cells impair macrophage-dependent ingestion of dying cells in a manner dependent on direct cell-to-cell contact. Manipulation of efferocytosis is shown rely on macrophage-intrinsic SIRPa-ITIM signalling via heightened CD47/24 expression by senescent cells. The translational implication of the study is provided by demonstrating that senescent cell inhibition of efferocytosis can be partially reversed by small molecules and antibodies targeting factors involved in "don't-eat-me" signalling.

Collectively, the study comprehensively and convincingly documents a role of senescent cell manipulation of macrophages by using complementary approaches involving cell biology and in vivo analysis. The paper will be of interest to the senescence, efferocytosis, and ageing fields. I have only minor suggestions for improvement.

- Fig. 4. To judge the impact of CD47 KO or manipulation a side-by-side comparison would be particularly useful to assess the effect of CD47 KO on SECS (Fig. 4B) or absence of senescent cells (Fig. 4E/F).
- Is CD24 also upregulated in senescent fibroblasts? The epithelial cell-specific usage of CD24 could be discussed.
- Page 11: Figure 5E-G instead of Figure E-G

Reviewer #3 (Comments to the Authors (Required)):

In aging, senescent cells often accumulate and are thought to have a detrimental impact on the health of the host. The present study from Schloesser et al. explores if and how senescent cells affect macrophage efferocytosis, a process essential for homeostasis that is also thought to be impaired in aging. The authors make the fascinating discovery that senescent cells suppress phagocytosis of apoptotic cells. Although the notion that senescent cells are "refractory" to phagocytosis is not surprising, per se, the authors provide compelling evidence that senescent cells are a potentially dangerous complication of aging by demonstrating that senescent cells suppress efferocytosis via a contact-mediated inhibition mechanism. The findings are interesting, the experiments are well-performed (especially the use of Fabs), and the implications are clear. However, this reviewer thinks that a few major points are worth addressing prior to acceptance. Additionally, the figures and figure captions feature some errors, including misused statistics and inaccurate reporting of n's that require fixing.

Major Points

1. Figure 1 (and generally). The data generated using incucyte is quite striking, however I find the time courses difficult to interpret given a typical efferocytosis assay is much shorter. I think it is important for the authors to supplement their current experiments with an approach to investigate per-cell engulfment rate. There are a few approaches that would be reasonable, for instance, co-labeling of targets with a pH-insensitive indicator (DAPI, TAMRA-SE) and pH-sensitive indicator (pHrodo, CypHer5E), and analyzing efferocytosis using either flow cytometry (MFI in actively-engulfing cells) or microscopy. I think it is especially important to know what is happening at earlier timepoints. Is rate of degradation affected? Are macrophages eating more or less per cell? Microscopy might also provide additional information about particulate uptake size, especially in senescence cell conditions.
2. It is unclear from the literature cited what the actual abundance of senescent cells are compared to, say, the number of apoptotic cells in a given tissue. I don't think this detracts from the authors' findings, but I think it is a worthwhile contribution for the authors to vary the ratios of senescent cells to apoptotic cells. To simplify the experiments, perhaps an interesting question is whether senescent cell burden differentially affects efferocytotic capacity? Similarly, does senescent cell burden affect CD47/CD24/SHP treatment efficacy?

Minor Point:

1. Most imaging experiments (both WB and microscopy) do not report how many independent experiments, scenes, frames, etc that images are representative of.
2. In Figure 4J, the caption reads data is from three independent experiments, however the figure displays either two or four dots, depending on condition. In Figure 4K, the caption states the data is from two independent experiments, whereas the figure shows 4 dots per condition. Is it two or four? If two, what do the dots represent? Additionally, if two replicates, this is an inappropriate use of a parametric statistic.
3. Figure 6C and 6H. The authors perform statistics on n=2, which is an inappropriate use of a parametric statistic. The magnitudes are quite strong and the error small, so statistics seem unnecessary for this figure, especially given the authors performed the experiments across several lines. I would recommend removing p values.
4. The authors state in the discussion "However, our experiments using CD47-deficient murine or human senescent fibroblasts clearly show that loss of CD47 is not sufficient to trigger macrophage engulfment and other signals independent of CD47 restrict "self" phagocytosis." However, this assumes that self-phagocytosis beyond the clearance of red blood cells (a cell that lacks typical apoptotic machinery) is a meaningful biological process. Is it not as likely that the reason senescent cell accumulation is dangerous is because they expose no signals for clearance (much like healthy cells) but, unlike healthy cells, expose signals that repress homeostatic efferocytosis?
5. The authors' discussion of previous studies reporting senescent cell clearance seems incomplete and should be strengthened. The authors have convinced this reviewer of their findings, but it seems reasonable to discuss potential differences in findings more evenhandedly.
6. Could the authors speculate in the discussion about what is driving the exposure of these signals in senescent cells?
7. The authors should soften their conclusions in the text in regard to Figure 4E and 4F given the lack of statistical significance.

Response to Reviewers

Overview: The main comments from the reviewers concerned clarifications and additions to the discussions. Reviewer 3 suggested several straightforward experiments to clarify some of the key points about the early events following a senescent cell encounter with macrophages; we were able to perform these experiments, which are detailed below. The revised manuscript has been formatted per the journal requirements; in this case, a slight re-ordering and re-numbering of the figures and supplemental information was necessary.

All changes in the manuscript text are highlighted in **yellow**. A detailed point-by-point response is provided below.

Reviewer #1

The authors state that: "Blocking CD47 (Figure 4D) on human senescent cells (primary lung fibroblasts derived from healthy (Figure 4E) or IPF-lungs (Figure 4F)) using anti-CD47 FAB fragments augmented corpse removal by co-cultured MDMs relative to co-cultures without CD47 blockade." This is not clear from the data presented, as there seems to be no significant differences in the AUC between untreated and anti-CD47 treated conditions.

Response: In the original version we noted a numerical increase but no statistically significant increase of corpse removal by adding anti-CD47 FAB fragment to the co-culture. However, we reviewed the raw data and identified an error (in figure preparation not the experimental data) when specifying the p-value. Instead of a p-value of 0.5, the actual p-value is 0.053. For clarity and transparency, we have included the raw data along with the calculations for the statistics, which is appended at the end of this document.

NSC87877 is both a potent SHP-1 and SHP-2 inhibitor. It would be important for the readers that this is clarified.

Response: We have this in the manuscript accordingly (page 11).

The manuscript is already very comprehensive. It would have been ideal to complement the SHP-1 immunolocalization studies with biochemical approaches on its association with SIRP α . A discussion on this would be useful for the readers as well.

Response: Co-immunolocalization of SIRP α and SHP-1 would indeed be ideal to show translocation of SHP1 to the cellular membrane after SIRP α activation. However, the anti-SIRP α antibodies we have used for immunofluorescence do not perform well, likely because the fixation of the cells alters the epitope. In the future, we have to screen (against the SIRP α knockout cells) for the best antibodies.

Reviewer #2

Fig. 4. To judge the impact of CD47 KO or manipulation a side-by-side comparison would be particularly useful to assess the effect of CD47 KO on SECS (Fig. 4B) or absence of senescent cells (Fig. 4E/F).

Response: We agree that a site by site comparison of senescent CD47 KO cells and manipulation (e.g., by anti-CD47 FABs) would be ideal. However, at present we cannot perform this experiment because the FAB reagents do not cross-react with murine cells. In the human setting when using Panc1 CD47 KO cells side by side with anti-CD47 FAB fragments, CD24 compensates in the absence of CD47.

Is CD24 also upregulated in senescent fibroblasts? The epithelial cell-specific usage of CD24 could be discussed.

Response: We respectfully refer the reviewer to Supplemental Figure 3C and page 10. Here, we showed by western blot analysis that CD24 is only detectable on senescent epithelial cells and not fibroblasts or liver stellate cells. Thus, epithelial cells utilize the CD24-Siglec10 axis in addition to the CD47-SIRP α axis seem to mediate SECS.

Page 11: Figure 5E-G instead of Figure E-G

Response: Thank you for the comment, we added the missing figure reference.

Reviewer #3

Major Points

1. Figure 1 (and generally). The data generated using IncuCyte is quite striking, however I find the time courses difficult to interpret given a typical efferocytosis assay is much shorter. I think it is important for the authors to supplement their current experiments with an approach to investigate per-cell engulfment rate. There are a few approaches that would be reasonable, for instance, co-labeling of targets with a pH-insensitive indicator (DAPI, TAMRA-SE) and pH-sensitive indicator (pHrodo, CypHer5E), and analyzing efferocytosis using either flow cytometry (MFI in actively-engulfing cells) or microscopy. I think it is especially important to know what is happening at earlier timepoints. Is rate of degradation affected? Are macrophages eating more or less per cell? Microscopy might also provide additional information about particulate uptake size, especially in senescence cell conditions.

Response: We agree with this point and it is a really good experiment. We exposed human macrophages co-cultured with either proliferating or senescent cells to apoptotic Raji cells co-labelled with a pH-insensitive dye (cell tracker green) and a pH-sensitive dye (CypHer5E). To investigate engulfment but also degradation of apoptotic corpses at early time points, we washed and fixed the co-cultures 1h, 2h or 4h post feeding to address the “early” events as suggested. Prior to confocal microscopy analysis we performed immunofluorescent staining for CD45 (to label all macrophages) and Hoechst (to label all nuclei). Our results indicate that at early time points, senescent cells impair macrophages’ ability to engulf apoptotic corpses while the acidification of the phagolysosome does not seem to be decreased.

These data support the basic conclusion of the manuscript: phagocytosis (via the CD47 pathway) is the main event targeted by senescent cells; once a macrophage engulfs a corpse, then digestion in the phagolysosome occurs normally. In the future, we intend to extend this line of experimentation by performing metabolomics of the macrophages and the cell supernatant, which would provide evidence that corpse digestion is normal once it occurs.

2. It is unclear from the literature cited what the actual abundance of senescent cells are compared to, say, the number of apoptotic cells in a given tissue. I don't think this detracts from the authors' findings, but I think it is a worthwhile contribution for the authors to vary the ratios of senescent cells to apoptotic cells. To simplify the experiments, perhaps an interesting question is whether senescent cell burden differentially affects efferocytotic capacity? Similarly, does senescent cell burden affect CD47/CD24/SHP treatment efficacy?

Response: The first part of this question cuts to heart of the main current theoretical and practical issue in the senescence field: how can senescent cells be accurately quantified, including their numbers relative to apoptotic corpses and all the other cells in a tissue microenvironment? There are many limitations associated with this issue: apoptotic cells can be stained (and in part tracked using fluorescent approaches) but this provides (in most cases) a snapshot of the dynamics of what happens

in vivo. Similarly, the first-generation senescent cell “reporters” such as the Sharpless p16 reporter mice did not provide a satisfactory answer to the quantification question. However, last week Reyes et al. published a new p16 reporter (which we cite) that seems to open the possibility of far higher fidelity in tracking senescent cells.

The second part of the reviewer’s question is also an important issue. In order to address the question of ratios, a vastly increased series of experimental studies is needed (using complementary techniques) with numerous comparisons of the different variables (senescent cells, macrophages, corpse numbers and CD47, CD24 and SHP-1 treatments. We do not discount the importance of this issue or the reviewer’s careful consideration of the question. However, at this point, our manuscript is already at the maximum number of figures (10) and the experimental series, to be sufficiently comprehensive and with high rigor, would take at least a year to perform properly and is thus beyond the scope of the present submission.

Minor Points:

1. Most imaging experiments (both WB and microscopy) do not report how many independent experiments, scenes, frames, etc that images are representative of.

Response: We have added the missing information in the figure legends.

2. In Figure 4J, the caption reads data is from three independent experiments, however the figure displays either two or four dots, depending on condition. In Figure 4K, the caption states the data is from two independent experiments, whereas the figure shows 4 dots per condition. Is it two or four? If two, what do the dots represent? Additionally, if two replicates, this is an inappropriate use of a parametric statistic.

Response: We changed the inaccuracies in the respective figures and figure legends.

3. Figure 6C and 6H. The authors perform statistics on n=2, which is an inappropriate use of a parametric statistic. The magnitudes are quite strong and the error small, so statistics seem unnecessary for this figure, especially given the authors performed the experiments across several lines. I would recommend removing p values.

Response: We removed the p-values. In Figure 6H (now Figure 10H), we increased the n-number to 4 independent experiments and performed statistics.

4. The authors state in the discussion "However, our experiments using CD47-deficient murine or human senescent fibroblasts clearly show that loss of CD47 is not sufficient to trigger macrophage engulfment and other signals independent of CD47 restrict "self" phagocytosis." However, this assumes that self-phagocytosis beyond the clearance of red blood cells (a cell that lacks typical apoptotic machinery) is a meaningful biological process. Is it not as likely that the reason senescent cell accumulation is dangerous is because they expose no signals for clearance (much like healthy cells) but, unlike healthy cells, expose signals that repress homeostatic efferocytosis?

Response: The reviewer raises an important point with this comment. At the present time, one of the main limitations in senescent cell research is *in vivo* identification of senescent cells as noted above, which would open the door to understanding their clearance, and how they regulate other cells in the tissue environment. The reviewer asks if self-phagocytosis *beyond* the clearance of red blood cells is a meaningful biological process: we do not have a good answer to this question one way or the other, which requires new technologies. The Reyes et al. paper that was recently published in *Science* (October 14, 2022 issue) describing a second generation p16 reporter is a major advance in this regard. To this end, the statement in the discussion has been modified including the addition of a key reference from Karin et al., which provides a thoughtful discussion of the “clearance vs. accumulation” issue.

5. The authors' discussion of previous studies reporting senescent cell clearance seems incomplete and should be strengthened. The authors have convinced this reviewer of their findings, but it seems reasonable to discuss potential differences in findings more evenhandedly.

Response: We agree on this point. The counter to this issue is very little is known about senescent cell clearance because of the limitation in definitively tracking a senescent cell *in vivo*, and tying its fate to – for example – an immunological process. Nevertheless, we added a statement to consider other work such as Scott Lowe's evidence for NK-mediated senescent cell removal. In discussing this issue frequently amongst the authors, we note that the time is right for a comprehensive review on this issue.

6. Could the authors speculate in the discussion about what is driving the exposure of these signals in senescent cells?

Response: We can only speculate about this issue at present and we feel it is important not to go too far. One possibility is combinations of inflammatory signals in local environments. Accordingly, we mentioned this in the discussion but only briefly.

7. The authors should soften their conclusions in the text in regard to Figure 4E and 4F given the lack of statistical significance.

Response: As already mentioned for the first comment of reviewer #1 (see above), we must also admit that we made a typing error when specifying the p-value. Instead of a p-value of 0.5, we have a p-value of 0.053, which means that a zero is missing in the manuscript. We agree that there is no significant increase of corpse removal if an anti-CD47 FAB fragment was added to the co-culture between macrophages and senescent IPF-derived fibroblasts. However, we think that the overall trend is there, which is close to statistical significance.

FAB fragment analysis for the reviewers:

IPF raw data: AUC [2-20h]

	Group A	Group B
	ctrl	anti-CD47 FAB (25µg/ml)
1	1.0407596e+007	1.9671250e+007
2	1.8822534e+007	2.1059554e+007
3	1.4339058e+007	2.6273238e+007
4		1.9188677e+007
5		

Results unpaired t-test:

Tabular results	
Unpaired t test Tabular results	
1	Table Analyzed
2	Fig 4F_IPF_AUC_CD47 FAB
3	Column B
4	anti-CD47 FAB (25µg/ml)
4	vs.
5	Column A
6	ctrl
7	Unpaired t test
8	P value
9	0.0538
9	P value summary
10	ns
10	Significantly different (P < 0.05)?
11	No
11	One- or two-tailed P value?
12	Two-tailed
12	t, df
13	t=2.511, df=5
14	How big is the difference?
15	Mean of column A
16	14523063
16	Mean of column B
17	21548180
17	Difference between means (B - A) ± SEM
18	7025117 ± 2798154
18	95% confidence interval
19	-167767 to 14218001
19	R squared (eta squared)
20	0.5576
21	F test to compare variances
22	F, DFn, Dfd
23	1.680, 2, 3
23	P value
24	0.6479
24	P value summary
25	ns
25	Significantly different (P < 0.05)?
26	No
27	Data analyzed
28	Sample size, column A
29	3
29	Sample size, column B
30	4

October 26, 2022

RE: JCB Manuscript #202207097R

Prof. Peter Murray
Max Planck Institute of Biochemistry
Am Klopferspitz 18
Martinsried 82152
Germany

Dear Prof. Murray:

Thank you for submitting your revised manuscript entitled "Senescent cells suppress macrophage-mediated corpse removal via upregulation of the CD47-QPCT/L axis". We have now assessed your revised manuscript and we would be happy to publish your paper in JCB pending final revisions necessary to meet our formatting guidelines (see details below).

To avoid unnecessary delays in the acceptance and publication of your paper, please read the following information carefully. Please go through all the formatting points paying special attention to those marked with asterisks.

A. MANUSCRIPT ORGANIZATION AND FORMATTING:

1) Text limits: Character count for Articles and Tools is < 40,000, not including spaces. Count includes title page, abstract, introduction, results, discussion, and acknowledgments. Count does not include materials and methods, figure legends, references, tables, or supplemental legends.

2) Figures limits: Articles and Tools may have up to 10 main text figures.

Please note that main text figures should be provided as individual, editable files.

3) Figure formatting:

Molecular weight or nucleic acid size markers must be included on all gel electrophoresis.

***** Scale bars must be present on all microscopy images, including inset magnifications. Please include scale bars in main Figs. 1A and 9B (inset magnifications).**

***** Also, please avoid pairing red and green for images and graphs to ensure legibility for color-blind readers. If red and green are paired for images, please ensure that the particular red and green hues used in micrographs (main Figs. 2B and 6D) are distinctive with any of the colorblind types. If not, please modify colors accordingly or provide separate images of the individual channels.**

4) Statistical analysis:

Error bars on graphic representations of numerical data must be clearly described in the figure legend.

The number of independent data points (n) represented in a graph must be indicated in the legend. Please, indicate whether N refers to technical or biological replicates (i.e. number of analyzed cells, samples or animals, number of independent experiments).

If independent experiments with multiple biological replicates have been performed, we recommend using distribution-reproducibility SuperPlots (please, see Lord et al., JCB 2020) to better display the distribution of the entire dataset, and report statistics (such as means, error bars, and P values) that address the reproducibility of the findings.

***** Statistical methods should be explained in full in the materials and methods in a separate section.**

For figures presenting pooled data the statistical measure should be defined in the figure legends.

Please also be sure to indicate the statistical tests used in each of your experiments (both in the figure legend itself and in a separate methods section) as well as the parameters of the test (for example, if you ran a t-test, please indicate if it was one- or two-sided, etc.).

*** As you used parametric tests in your study (i.e. t-tests), you should have first determined whether the data was normally distributed before selecting that test. In the stats section of the methods, please indicate how you tested for normality. If you did not test for normality, you must state something to the effect that "Data distribution was assumed to be normal but this was not formally tested."

5) Abstract and title:

The abstract should be no longer than 160 words and should communicate the significance of the paper for a general audience.

The title should be less than 100 characters including spaces. Make the title concise but accessible to a general readership.

6) Materials and methods:

Should be comprehensive and not simply reference a previous publication for details on how an experiment was performed. The text should not refer to methods "...as previously described."

Also, the materials and methods should be included with the main manuscript text and not in the supplementary materials.

7) Please be sure to provide the sequences for all your primers/oligos and RNAi constructs in the materials and methods.

*** You must also indicate in the methods the source, species, and catalog numbers (where appropriate) for all your antibodies. Please include species for all your antibodies.

8) Microscope image acquisition:

The following information must be provided about the acquisition and processing of images:

- a. Make and model of microscope
- b. Type, magnification, and numerical aperture of the objective lenses
- c. Temperature
- d. imaging medium
- e. Fluorochromes
- f. Camera make and model
- g. Acquisition software
- h. Any software used for image processing subsequent to data acquisition. Please include details and types of operations involved (e.g., type of deconvolution, 3D reconstitutions, surface or volume rendering, gamma adjustments, etc.).

10) Supplemental materials:

There are strict limits on the allowable amount of supplemental data. Articles/Tools may have up to 5 supplemental figures. There is no limit for supplemental tables.

Please note that supplemental figures and tables should be provided as individual, editable files.

*** A summary of all supplemental material should appear at the end of the Materials and Methods section (please see any recent JCB paper for an example of this summary).

11) eTOC summary:

A ~40-50 word summary that describes the context and significance of the findings for a general readership should be included on the title page.

The statement should be written in the present tense and refer to the work in the third person. It should begin with "First author name(s) et al..." to match our preferred style.

12) Conflict of interest statement:

JCB requires inclusion of a statement in the acknowledgements regarding competing financial interests. If no competing financial interests exist, please include the following statement: "The authors declare no competing financial interests."

13) A separate author contribution section is required following the Acknowledgments in all research manuscripts.

*** All authors should be mentioned and designated by their first and middle initials and full surnames and the CRediT nomenclature should be used (<https://casrai.org/credit/>).

14) ORCID IDs: ORCID IDs are unique identifiers allowing researchers to create a record of their various scholarly contributions in a single place. At resubmission of your final files, please consider providing an ORCID ID for as many contributing authors as possible.

15) Materials and data sharing:

All animal and human studies must be conducted in compliance with relevant local guidelines, such as the US Department of Health and Human Services Guide for the Care and Use of Laboratory Animals or MRC guidelines, and must be approved by the authors' Institutional Review Board(s). A statement to this effect with the name of the approving IRB(s) must be included in the Materials and Methods section.

*** As a condition of publication, authors must make protocols and unique materials (including, but not limited to, cloned DNAs; antibodies; bacterial, animal, or plant cells; and viruses) described in our published articles freely available upon request by researchers, who may use them in their own laboratory only. All materials must be made available on request and without undue delay. Please, indicate whether the mice strains, cell lines and reagents generated in this study have been deposited in public repositories. If not, please state that they would be made available to the scientific community upon request in the 'Data availability' section.

*** All datasets included in the manuscript must be available from the date of online publication, and the source code for all custom computational methods, apart from commercial software programs, must be made available either in a publicly available database or as supplemental materials hosted on the journal website. Numerous resources exist for data storage and sharing (see Data Deposition: <https://rupress.org/jcb/pages/data-deposition>), and you should choose the most appropriate venue based on your data type and/or community standard. If no appropriate specific database exists, please deposit your data to an appropriate publicly available database. Please, deposit your proteomics data in a proper public repository and include the accession number in the Methods section.

16) Please note that JCB now requires authors to submit Source Data used to generate figures containing gels and Western blots with all revised manuscripts. This Source Data consists of fully uncropped and unprocessed images for each gel/blot displayed in the main and supplemental figures. The Source Data files will be directly linked to specific figures in the published article.

Since your paper includes cropped gel and/or blot images, please be sure to provide one Source Data file for each figure that contains gels and/or blots along with your revised manuscript files. File names for Source Data figures should be alphanumeric without any spaces or special characters (i.e., SourceDataF#, where F# refers to the associated main figure number or SourceDataFS# for those associated with Supplementary figures). The lanes of the gels/blots should be labeled as they are in the associated figure, the place where cropping was applied should be marked (with a box), and molecular weight/size standards should be labeled wherever possible.

B. FINAL FILES:

Thank you for this interesting contribution, we look forward to publishing your paper in Journal of Cell Biology.

Sincerely,

Jennifer Stow
Monitoring Editor
Journal of Cell Biology

Lucia Morgado-Palacin, PhD
Scientific Editor
Journal of Cell Biology